# Feature Importance Ranking for Deep Learning

**Maksymilian A. Wojtas**     **Ke Chen**
Department of Computer Science, The University of Manchester, Manchester M13 9PL, U.K.
`{maksymilian.wojtas,ke.chen}@manchester.ac.uk`

## Abstract

Feature importance ranking has become a powerful tool for explainable AI. How-ever, its nature of combinatorial optimization poses a great challenge for deep learning. In this paper, we propose a novel dual-net architecture consisting of operator and selector for discovery of an optimal feature subset of a fixed size and ranking the importance of those features in the optimal subset simultaneously. During learning, the operator is trained for a supervised learning task via optimal feature subset candidates generated by the selector that learns predicting the learn-ing performance of the operator working on different optimal subset candidates. We develop an alternate learning algorithm that trains two nets jointly and incorpo-rates a stochastic local search procedure into learning to address the combinatorial optimization challenge. In deployment, the selector generates an optimal feature subset and ranks feature importance, while the operator makes predictions based on the optimal subset for test data. A thorough evaluation on synthetic, benchmark and real data sets suggests that our approach outperforms several state-of-the-art feature importance ranking and supervised feature selection methods. (Our source code is available: https://github.com/maksym33/FeatureImportanceDL)

## 1 Introduction

In machine learning, feature importance ranking (FIR) refers to a task that measures contributions of individual input features (variables) to the performance of a supervised learning model. FIR has become one of powerful tools in explainable/interpretable AI [1] to facilitate understanding of decision-making by a learning system and discovery of key factors in a specific domain, e.g., in medicine, what genes are likely main causes of a cancer [2].

Due to the existence of correlated/dependent and irrelevant features to targets in high-dimensional real data, feature selection [3] is often employed to address the well-known curse of dimensionality challenge and to improve the generalization of a learning system, where a subset of optimal features is selected in terms of the pre-defined criteria to maximize the performance of a learning system. Feature selection may be conducted at either population or instance level; the populationwise methods would find out an optimal feature subset collectively for all the instances in a population, while the instancewise ones tend to uncover a subset of salient features specific to a single instance. In practice, FIR is always closely associated with feature selection by ranking the importance of those features in an optimal subset and can also be used as a proxy for feature selection, e.g., [2, 4, 5].

Deep learning has turned out to be extremely powerful in intelligent system development but its purported "black box" nature makes it extremely difficult to be applied to tasks demanding explain-ability/interpretability. Recently, FIR for deep learning has become an active research area where most works focus on instancewise FIR [6] and only few works exist for populationwise FIR/feature selection, e.g., [7]. In a populationwise scenario, feature selection needs to find an optimum in detecting any functional dependence between input data and targets, which is NP-hard in general [8]. High degree of nonlinearity in deep learning execrates this combinatorial optimization problem.

In this paper, we address a populationwise FIR issue in deep learning: for a feature set, finding an optimal feature subset of a fixed size that maximizes the performance of a deep neural network and ranking the importance of all the features in this optimal subset simultaneously. To tackle this problem, we propose a novel dual-net neural architecture, where an operator net works for a supervised learning task via optimal subset candidates provided by a selector net that learns finding the optimal feature subset and ranking feature importance via the learning performance feedback of the operator. Two nets are jointly trained in an alternate manner. After learning, the selector net is used to find an optimal feature subset and rank feature importance, while the operator net makes predictions based on the optimal feature subset for test data. A thorough evaluation on synthetic, benchmark and real datasets via a comparative study manifests that our approach leveraged by deep learning outperforms several state-of-the-art FIR and supervised feature selection methods.

## 2 Related Work

In the context of deep learning, there exist three methods for FIR; i.e., *regularization*, *greedy search* and *averaged input gradient*. The deep feature selection (DFS) [7] was proposed for FIR with the same idea behind the regularized linear models [9, 10]. The DFS suffers from several issues, e.g. a high computational burden in finding an optimal regularization hyper-parameters and vanishing gradient. Moreover, the weight-shrinkage idea [9, 10] may not always work for complex dependence between input features and targets since the use of shrunk weights as feature importance is theoretically justifiable to linear models only. It seems straightforward to apply a greedy search method, e.g., forward subset selection (FS) [11], to deep learning for FIR. Obviously, this method inevitably incurs extremely high computational cost and may end up with only a sub-optimal result. Finally, some instancewise FIR methods have been converted into populationwise ones, e.g., the averaged input gradient (AvGrad) [12] that uses the average of all the saliency maps extracted from individual instances for FIR and global aggregation [13, 14, 15] that uses different aggregation mechanisms to achieve the populationwise feature importance ranking. As local explanations are specific at the instance level and often inconsistent with global explanations at the population level, the simple accumulation of instancewise FIR results may not work on populationwise FIR. In contrast, our method would overcome all the limitations stated above.

In machine learning, regularized linear models, e.g., LASSO [9], and random forest (RF) [16] are two off-the-shelf FIR methods. Other strong FIR methods include the SVM-based RFE [2] and the dependence-maximization based BAHSIC [4, 5]. In general, such methods may have the limited learning capacity for complex tasks in comparison to deep learning, and may not always work for complex dependence between input features and targets. On the other hand, according to the definition in [4, 17], our FIR problem formulation can be treated a sub-problem of supervised feature selection when the size of an optimal feature subset is pre-specified. To this end, our method is closely related to several strong feature selection methods with the same setting, including those working on mutual information criteria, e.g., mRMR [18] and the kernel-based CCM [17] although such methods do not consider FIR. Leveraged with deep learning, our approach is more effective than those aforementioned FIR and supervised feature selection methods, as manifested in our experiments.

## 3 Method

### 3.1 Problem Formulation

Suppose $\mathcal{D} = \{\mathcal{X}, \mathcal{Y}\}$ is a dataset used for supervised learning. In this data set, $(\boldsymbol{x}, \boldsymbol{y})$ is a training example, where $\boldsymbol{x} \in \mathcal{X}$ is a vector of $d$ features and $\boldsymbol{y} \in \mathcal{Y}$ is its corresponding target. Let $\boldsymbol{m} \in \mathcal{M}$ denote a $d$-dimensional binary mask vector of $0/1$ elements, where $||\boldsymbol{m}||_0 = s$, $s < d$ and $|\mathcal{M}| = \binom{d}{s}$. Thus, we can use such a mask vector to indicate a feature subset: $\{\boldsymbol{x} \otimes \boldsymbol{m}\}_{\boldsymbol{x} \in \mathcal{X}}$, where $\otimes$ denotes Hadamard product that yield a subset of $s$ features for any instance $\boldsymbol{x} \in \mathcal{X}$. Assume that $\mathcal{Q}(\boldsymbol{x}, \boldsymbol{m})$ quantifies the instance-level performance of a learning system trained on $\mathcal{D}$ via a feature subset, $\{\boldsymbol{x} \otimes \boldsymbol{m}\}_{\boldsymbol{x} \in \mathcal{X}}$, the *feature importance ranking* (FIR) can then be formulated as follows:

$$\big(\boldsymbol{m}^*, \text{Score}(\boldsymbol{m}^*)\big) = \underset{\boldsymbol{m} \in \mathcal{M}}{\text{argmax}} \sum_{\boldsymbol{x} \in \mathcal{X}} \mathcal{Q}(\boldsymbol{x}, \boldsymbol{m}), \tag{1}$$

where $\boldsymbol{m}^*$ is the indicator of an optimal feature subset discovered by an FIR algorithm and $\text{Score}(\boldsymbol{m}^*)$ quantifies the importance of all the selected features in this optimal subset.

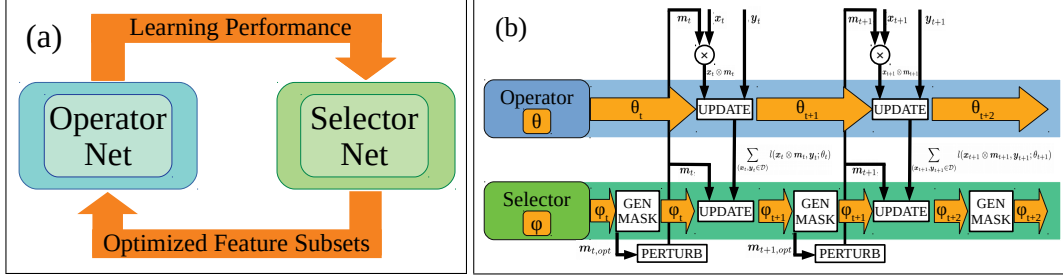

Figure 1: Our feature importance ranking model. (a) Dual-net architecture. (b) Parameter update.

Ideally, an FIR approach should be able to: 1) detect any functional dependence between input features and targets; 2) rank the importance of all the selected features to reflect their contributions to the learning performance; and 3) preserve the detected functional dependence and the feature importance ranking in test data.

## 3.2 Model Description

To tackle the FIR problem stated in Eq.(1) effectively with three criteria described in Sect. 3.1, we propose a deep learning model of dual nets, *operator* and *selector*, as shown in Fig. 1(a). The operator net is employed to accomplish a supervised learning task, e.g., classification or regression, on a given feature subset provided by the selector net, while the selector net is designated to learn finding out an optimal feature subset based on the performance feedback of the operator net working on optimal feature subset candidates during learning. Both the operator and the selector nets are trained jointly in an alternate manner (c.f. Sect. 3.3) to reach a synergy for the FIR.

Technically, the operator is carried out with a deep neural network parameterized with $\theta$, $f_O(\theta; \boldsymbol{x}, \boldsymbol{m})$, for a given task, e.g., multi-layer perceptron (MLP) or convolution neural network (CNN). This net is trained on $\mathcal{D}$ based on different feature subsets to learn $f_O : \mathcal{X} \times \mathcal{M} \to \mathcal{Y}$. After learning (c.f. Sect. 3.3), the trained operator net, $f_O(\theta^*; \boldsymbol{x}, \boldsymbol{m}^*)$, is applied to the test data for prediction, where $\theta^*$ is the optimal parameters of the operator net and $\boldsymbol{m}^*$ is generated by the trained selector net (c.f. Sect. 3.4).

In our method, the selector is implemented with an MLP parameterized with $\varphi$, $f_S(\varphi; \boldsymbol{m})$. As defined in Eq.(1), a selected optimal feature subset should maximize the averaging performance of the operator quantified by $\mathcal{Q}(\boldsymbol{x}, \boldsymbol{m})$ for all $\boldsymbol{x} \in \mathcal{X}$. Thus, we want the selector net to learn predicting the averaging performance of the operator net on different feature subsets; i.e., $f_S : \mathcal{M} \to \mathbb{R}$. After being trained properly (c.f. Sect. 3.3), we can use an algorithm working on the trained selector net of the optimal parameters $\varphi^*$, $f_S(\varphi^*; \boldsymbol{m})$, to generate an optimal feature subset indicated by $\boldsymbol{m}^*$ and rank feature importance to achieve $\mathrm{Score}(\boldsymbol{m}^*)$ (c.f. Sect. 3.4).

## 3.3 Learning Algorithm

In essence, the FIR defined in Eq.(1) is a combinatorial optimization problem. According to the no free lunch theory for optimization [19], no algorithm can perform better than a random strategy in expectation in the setting of combinatorial optimization. Therefore, our learning algorithm is developed by leveraging learning with a stochastic local search procedure enhanced by injecting noise [20] on a small number of candidate feature subsets, $\mathcal{M}' \subset \mathcal{M}$, to avoid the exhaustive search.

For a training data set, $\mathcal{D} = \{\mathcal{X}, \mathcal{Y}\} = \big\{(\boldsymbol{x}, \boldsymbol{y})\big\}_{\boldsymbol{x} \in \mathcal{X}, \boldsymbol{y} \in \mathcal{Y}}$, a mask subset, $\mathcal{M}'$, converts each training example $(\boldsymbol{x}, \boldsymbol{y}) \in \mathcal{D}$ into $|\mathcal{M}'|$ examples: $\big\{(\boldsymbol{x} \otimes \boldsymbol{m}, \boldsymbol{y})\big\}_{\boldsymbol{m} \in \mathcal{M}'}$. Thus, the loss functions on $\mathcal{M}'$ (changing during learning) for the operator and the selector nets are defined respectively as follows:

$$\mathcal{L}_O(\mathcal{D}, \mathcal{M}'; \theta) = \frac{1}{|\mathcal{M}'||\mathcal{D}|} \sum_{\boldsymbol{m} \in \mathcal{M}'} \sum_{(\boldsymbol{x}, \boldsymbol{y}) \in \mathcal{D}} l(\boldsymbol{x} \otimes \boldsymbol{m}, \boldsymbol{y}; \theta), \tag{2a}$$

$$\mathcal{L}_S(\mathcal{M}'; \varphi) = \frac{1}{2|\mathcal{M}'|} \sum_{\boldsymbol{m} \in \mathcal{M}'} \left( f_S(\varphi; \boldsymbol{m}) - \frac{1}{|\mathcal{D}|} \sum_{(\boldsymbol{x}, \boldsymbol{y}) \in \mathcal{D}} l(\boldsymbol{x} \otimes \boldsymbol{m}, \boldsymbol{y}; \theta) \right)^2. \tag{2b}$$

Here, $l(\boldsymbol{x} \otimes \boldsymbol{m}, \boldsymbol{y}; \theta)$ is an instance-level cross-entropy/categorical cross-entropy loss for binary/multi-class classification or the mean square error (MSE) loss for regression. In Eq.(2b), we utilize the loss of the operator net, $l(\boldsymbol{x} \otimes \boldsymbol{m}, \boldsymbol{y}; \theta)$, to characterize its learning performance, $\mathcal{Q}(\boldsymbol{x}, \boldsymbol{m})$, since maximizing $\mathcal{Q}(\boldsymbol{x}, \boldsymbol{m})$ is equivalent to minimizing $l(\boldsymbol{x} \otimes \boldsymbol{m}, \boldsymbol{y}; \theta)$. As described in Sect. 3.2, during learning, the operator net relies on the selector net to provide an optimal subset of marks, $\mathcal{M}'$, indicating different optimal feature subset candidates, while the selector net requires the performance feedback from the operator net, $l(\boldsymbol{x} \otimes \boldsymbol{m}, \boldsymbol{y}; \theta)$ for all $\boldsymbol{m} \in \mathcal{M}'$. Two nets in our learning model hence have to be trained alternately. Below, we present the main learning steps in our learning algorithm of two phases, while the pseudo code can be found from Sect. D in supplementary materials.

**Phase I: Initial Operator Learning via Exploration.** From the scratch, we start training the operator net by using a small number of random feature subsets for several epochs until it can yield the different performance on different feature subsets stably. Technically, in each epoch, we randomly draw a subset of different masks, $\mathcal{M}'_1$, from $\mathcal{M}$; i.e., $\mathcal{M}'_1 = \{\boldsymbol{m}_i | \boldsymbol{m}_i = \mathrm{Random}(\mathcal{M}, s)\}_{i=1}^{|\mathcal{M}'|}$, where $\mathrm{Random}(\mathcal{M}, s)$ is a function that randomly draws a $d$-dimensional mask of $s$ one-elements and $d-s$ zero-elements from $\mathcal{M}$. If $\theta$ is trained by stochastic gradient decent (SGD), then it is updated by $\theta'' \triangleq \theta' - \eta \nabla_\theta \mathcal{L}_O(\mathcal{D}, \mathcal{M}'_1; \theta)|_{\theta = \theta'}$[1] where $\eta$ is a learning rate. After $E_1$ epochs, we set $\theta_1 = \theta''(E_1)$ and $\boldsymbol{m}'_{1,opt} = \mathrm{argmin}_{\boldsymbol{m} \in \mathcal{M}'_1} \sum_{(\boldsymbol{x}, \boldsymbol{y}) \in \mathcal{D}} l(\boldsymbol{x} \otimes \boldsymbol{m}, \boldsymbol{y}; \theta_1)$ to be used at the beginning of Phase II-A; i.e., $t = 1$ as shown in Fig. 1(b).

**Phase II-A: Selector Learning via Operator's Feedback.** As illustrated in Fig. 1(b), the operator provides training examples for the selector at step $t$: $\left\{ \left(\boldsymbol{m}, \frac{1}{|\mathcal{D}|} \sum_{(\boldsymbol{x}, \boldsymbol{y}) \in \mathcal{D}} l(\boldsymbol{x} \otimes \boldsymbol{m}, \boldsymbol{y}; \theta_t)\right) \right\}_{\boldsymbol{m} \in \mathcal{M}'_t}$. By using the SGD with initializing $\varphi_1$ randomly, the parameters in the selector net, $\varphi$, are updated by $\varphi_{t+1} \triangleq \varphi_t - \eta \nabla_\varphi \mathcal{L}_S(\mathcal{M}'_t; \varphi)|_{\varphi = \varphi_t}$. Then, we adopt an *exploration-exploitation* strategy to generate a new mask subset, $\mathcal{M}'_{t+1}$, for the operator learning at step $t+1$. Thus, $\mathcal{M}'_{t+1}$ is divided into two mutually exclusive subsets: $\mathcal{M}'_{t+1} = \mathcal{M}'_{t+1,1} \cup \mathcal{M}'_{t+1,2}$. Motivated by the role of noise in stochastic local search [20], $\mathcal{M}'_{t+1,1}$ is generated via exploration to avoid overfitting: $\mathcal{M}'_{t+1,1} = \{\boldsymbol{m}_i | \boldsymbol{m}_i = \mathrm{Random}(\mathcal{M}, s)\}_{i=1}^{|\mathcal{M}'_{t+1,1}|}$. Motivated by the input gradient idea [12], $\mathcal{M}'_{t+1,2}$ is generated by exploitation of the selector net, $f_S(\varphi_{t+1}; \boldsymbol{m})$, as follows: **a) Generation of an optimal subset**. Starting with $d$-dimensional $\boldsymbol{m}_0 = \left(\frac{1}{2}, \cdots, \frac{1}{2}\right)$, meaning that every feature has the equal chance to be selected, we have $\boldsymbol{\delta}_{\boldsymbol{m}_0} = \frac{\partial f_S(\varphi_{t+1}; \boldsymbol{m})}{\partial \boldsymbol{m}}|_{\boldsymbol{m} = \boldsymbol{m}_0}$. As input features of the larger gradients contribute more to the learning performance of the operator, we can find top $s$ features based on their gradients by $(\boldsymbol{m}_{opt}, \bar{\boldsymbol{m}}_{opt}) = \mathrm{argsort}(\boldsymbol{\delta}_{\boldsymbol{m}_0}, s)$ where $\boldsymbol{m}_{opt}$ is the mask to indicate top $s$ features and $\bar{\boldsymbol{m}}_{opt}$ is the mask for the remaining $d-s$ features. To ensure the optimality of $\boldsymbol{m}_{opt}$, we come up with a three-step *validation* procedure: **i)** Re-evaluate the contributions of top $s$ features by $(\boldsymbol{m}_{opt}, \bar{\boldsymbol{m}}_{opt}) = \mathrm{argsort}(\boldsymbol{\delta}_{\boldsymbol{m}_{opt}}, s)$ where $\boldsymbol{\delta}_{\boldsymbol{m}_{opt}} = \frac{\partial f_S(\varphi_{t+1}; \boldsymbol{m})}{\partial \boldsymbol{m}}|_{\boldsymbol{m} = \boldsymbol{m}_{opt}}$; **ii)** Replace a feature of negative gradient in $\boldsymbol{m}_{opt}$ with the one of the largest gradient in $\bar{\boldsymbol{m}}_{opt}$ if there exists; **iii)** Further check the optimality via a function $(\boldsymbol{m}'_{opt}, \bar{\boldsymbol{m}}'_{opt}) = \mathrm{swap}(\boldsymbol{m}_{opt}, \bar{\boldsymbol{m}}_{opt})$ that yields $\boldsymbol{m}'_{opt}$ by swapping between the feature of least gradient in $\boldsymbol{m}_{opt}$ and the one of the largest gradient in $\bar{\boldsymbol{m}}_{opt}$. Repeat (i)-(iii) until $f_S(\varphi_{t+1}; \boldsymbol{m}_{opt}) \leq f_S(\varphi_{t+1}; \boldsymbol{m}'_{opt})$. After going through the validation procedure, $\boldsymbol{m}_{t+1,opt}$ is obtained for step $t+1$. **b) Generation of optimal subset candidates via perturbation**. As the optimal subset $\boldsymbol{m}_{t+1,opt}$ might be a local optimum, we would further inject noise to generate more optimal subset candidates by a perturbation function $\mathrm{Perturb}(\boldsymbol{m}_{opt}, s_p)$. For $s_p < s$, $\mathrm{Perturb}(\boldsymbol{m}_{opt}, s_p)$ randomly flips $s_p$ different elements in $\boldsymbol{m}_{opt}/\bar{\boldsymbol{m}}_{opt}$ from 1/0 to 0/1 and swaps between changed elements in $\boldsymbol{m}_{opt}$ and $\bar{\boldsymbol{m}}_{opt}$. Applying $\mathrm{Perturb}(\boldsymbol{m}_{opt}, s_p)$ repeatedly leads to multiple optimal subset candidates; **c) Formation of optimal subset candidates**. Assembling **a)** and **b)** leads to $\mathcal{M}'_{t+1,2} = \{\boldsymbol{m}_{t,best}\} \cup \{\boldsymbol{m}_{t+1,opt}\} \cup \{\boldsymbol{m}_i | \boldsymbol{m}_i = \mathrm{Perturb}(\boldsymbol{m}_{t+1,opt}, s_p)\}_{i=1}^{|\mathcal{M}'_{t+1,2}|-2}$. Here, we always include $\boldsymbol{m}_{t,best}$, the subset that leads to the best learning performance of the operator net in the last step (step $t$), as the most important subset candidate in the current step (step $t+1$) in order to make the operator learning progress steadily. Note that $\boldsymbol{m}_{t,best}$ may not be $\boldsymbol{m}_{t,opt}$.

**Phase II-B: Operator Learning via Optimal Subset Candidates from Selector.** After completing the training of Phase II-A at step $t$, the selector net provides the optimal subset candidates,

$\mathcal{M}'_{t+1} = \mathcal{M}'_{t+1,1} \cup \mathcal{M}'_{t+1,2}$, for the operator net, as illustrated in Fig. 1(b). At step $t+1$, the operator net is thus trained based on $\mathcal{M}'_{t+1}$ with SGD: $\theta_{t+1} \triangleq \theta_t - \eta \nabla_\theta \mathcal{L}_O(\mathcal{D}, \mathcal{M}'_{t+1}; \theta)|_{\theta=\theta_t}$.

As shown in Fig. 1, our alternate algorithm enables the operator and the selector nets to be trained jointly in Phase II until a pre-specified stopping condition is satisfied.

### 3.4 Deployment

After the learning described in Sect. 3.3 is accomplished, we obtain the optimal parameters of the operator and the selector nets, $\theta^*$ and $\varphi^*$.

By using the trained selector net, $f_S(\varphi^*; \boldsymbol{m})$, we find out an optimal feature subset with the same procedure used in Phase II-A as follows: 1) starting with $\boldsymbol{m}_0 = \left(\frac{1}{2}, \cdots, \frac{1}{2}\right)$, calculate the gradient $\boldsymbol{\delta}_{\boldsymbol{m}_0} = \frac{\partial f_S(\varphi^*; \boldsymbol{m})}{\partial \boldsymbol{m}}|_{\boldsymbol{m}=\boldsymbol{m}_0}$; 2) finding top $s$ features by $(\boldsymbol{m}^*, \bar{\boldsymbol{m}}^*) = \operatorname{argsort}(\boldsymbol{\delta}_{\boldsymbol{m}_0}, s)$, where $\boldsymbol{m}^*$ indicates the optimal subset of top $s$ features; and 3) going through the validation procedure described in Phase II.A to ensure the optimality of $\boldsymbol{m}^*$. Thus, feature importance ranking on the final $\boldsymbol{m}^*$ is done by setting $\operatorname{Score}(\boldsymbol{m}^*) = \frac{\partial f_S(\varphi^*; \boldsymbol{m})}{\partial \boldsymbol{m}}|_{\boldsymbol{m}=\boldsymbol{m}^*}$ and sorting the input gradients of selected features.

During test, for a test instance, $\hat{\boldsymbol{x}}$, the trained operator net, $f_O(\theta^*; \boldsymbol{x}, \boldsymbol{m})$, can be used to make a prediction, $f_O(\theta^*; \hat{\boldsymbol{x}}, \boldsymbol{m}^*)$, via $\hat{\boldsymbol{x}} \otimes \boldsymbol{m}^*$, which allows a supervised learning task to be done based on only the optimal feature subset, $\boldsymbol{m}^*$, found out with our proposed approach.

## 4 Experiments

In this section, we evaluate our approach on synthetic, benchmark and real-world datasets where we always use 5-fold cross-validation for evaluation and report the performance statistics, i.e., *mean* and *standard deviation* estimated on 5 folds. We describe our main settings briefly in the main text, and the details of all the experimental settings can be found from Sect. A in Supplementary Materials.

### 4.1 Synthetic Data

Our first evaluation employs 3 synthetic datasets in literature [17, 11] for feature selection regarding regression and binary/multiclass classification as follows:

**XOR as 4-way classification** [17]. Group 8 corners of the cube, $(v_0, v_1, v_2) \in \{-1, +1\}^3$, by the tuples $(v_0 v_2, v_1 v_2)$, leading to 4 sets of vectors paired with their negations $\{v^{(c)}, -v^{(c)}\}$. For a class $c$, a point is generated from the mixture distribution: $\frac{1}{2}[N(v^{(c)}, 0.5I_3) + N(-v^{(c)}, 0.5I_3)]$. Then, form a 10-D feature vector for each example by adding 7 standard noise features, $(X_3, \cdots, X_9) \sim N(0, I_7)$.

**Nonlinear regression** [17]. $Y = -2\sin(2X_0) + \max(X_1, 0) + X_2 + \exp(-X_3) + \epsilon$, where $(X_0, \cdots, X_9) \sim N(0, I_{10})$ and $\epsilon \sim N(0, 1)$, leading to a 10-D feature vector for each example.

**Binary classification** [11]. To generate examples, set $Y = -1$ when $(X_0, \cdots, X_9) \sim N(0, I_{10})$ and $Y = +1$ when $X_0$ through $X_3$ are standard normal conditioned on $9 \le \sum_{i=0}^{3} X_i^2 \le 16$ and $(X_4, \cdots, X_9) \sim N(0, I_6)$, resulting in a 10-D feature vector for each example.

For each dataset, we randomly generate 512 and 1024 examples, respectively, for training and test. With our problem formulation described in Sect. 3.1, our experiment on synthetic data simulates an application scenario that selects $s$ out of $d$ features where $s$ is larger than the number of features relevant to the target in a dataset. As there are up to 4 relevant features in the above 3 datasets, we choose $s = 5$ in our experiment and compare with all the methods reviewed in Sect. 2, including DFS [7], AvGrad [12], FS [11] based on MLP, LASSO [9], RF [16], RFE [2], BAHSIC [4, 5], mRMR [18] and CCM [17]. According to a taxonomy [3], DFS, AvGrad, RF and ours are embedding methods, FS is a wrapper method and all the others are filtering methods. For those filtering methods, we use the exactly same kernel SVM/SVR described in those papers [2, 4, 18, 5, 17] and an MLP on LASSO for classification/regression. While DFS, AvGrad, LASSO and RF work on FIR for all 10 features, all other methods work with the same setting as ours by finding out top 5 features and FIR.

Fig. 2 shows the feature selection and FIR results yielded by different methods regarding top 5 features on 3 synthetic datasets where the FIR scores are normalized in each method and the equal FIR score is set to all the features selected by those methods without considering FIR. It is observed

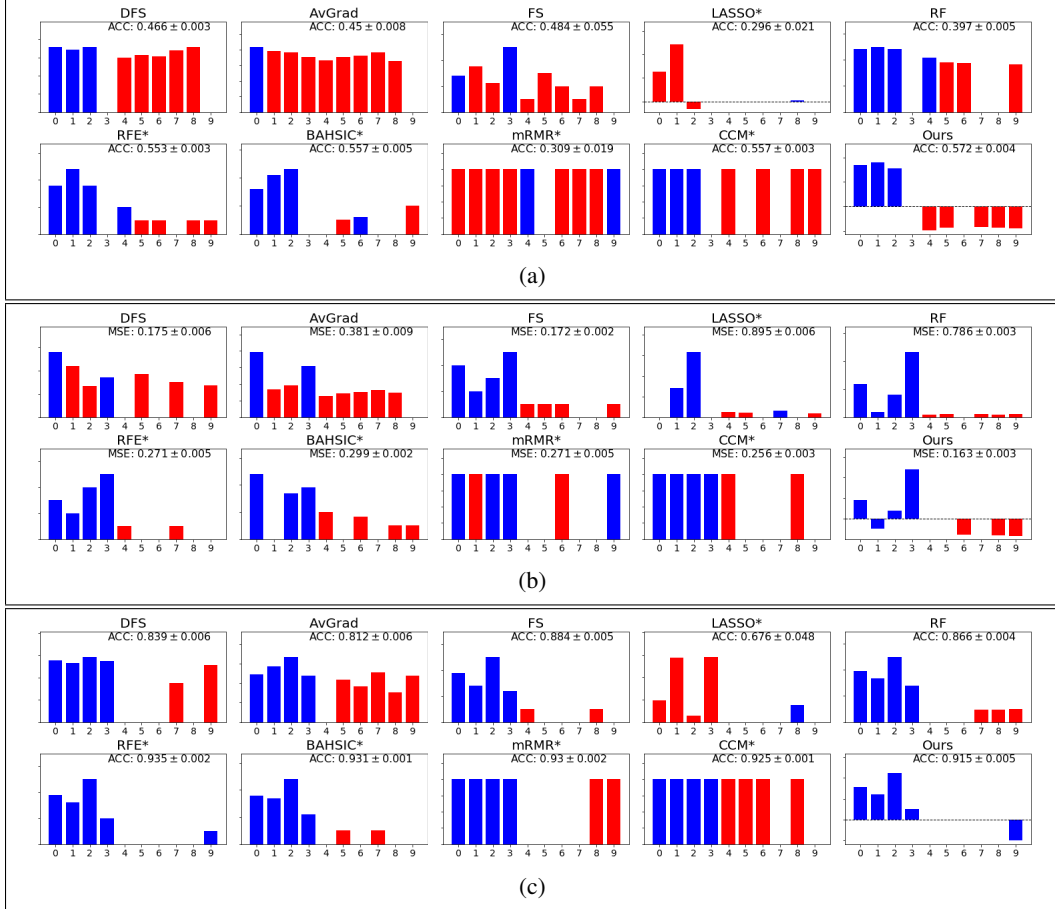

Figure 2: 5-fold cross-validation results (mean±str) on synthetic datasets ($s = 5, d = 10$). (a) XOR 4-way classification. (b) Nonlinear regression. (c) Binary classification. * refers to a filtering method, and blue/red colors indicate a feature selected in all 5 folds/fewer folds, respectively.

from Fig. 2 that our approach always finds out those relevant features in all 5 folds and does FIR properly by assigning negative scores (gradients), meaning unimportant, to irrelevant features. For the 4-way classification, DFS, RF, RFE, BAHSIC and CCM also find 3 relevant features in all 5 folds but others fail as shown in Fig. 2(a) although mRMR and CCM cannot yield FIR scores. In terms of accuracy, ours outperforms all other 9 methods despite the fact that DFS, AvGrad and RF work directly on the full feature set. For the nonlinear regression, FS, RF, RFE and CCM also select 4 relevant features in all 5 folds but ours yields the least MSE as shown in Fig. 2(b). For the binary classification, all the methods apart from LASSO find 4 relevant features in all 5 folds, as shown in Fig. 2(c). For this dataset, those state-of-the-art filtering methods yield better accuracy than others and the accuracy resulting from ours is slightly worse but comparable to those. In terms of FIR on all relevant features, ours is entirely consistent with those yielded by RF but performs significantly better than RF on 3 datasets. In comparison to the existing FIR methods for deep learning, ours always outperforms DFS, AvGrad and FS on 3 datasets in terms of both FIR and learning performance.

## 4.2 Benchmark Data

We further evaluate our approach on several well-known benchmark datasets from two different perspectives; i.e., explainability of FIR and learning performance on supervised feature selection. Evaluation on more benchmark datasets can be seen from Sect. B in Supplementary Materials.

**MNIST Dataset** [21]. To demonstrate the explainability of FIR via visual inspection, we employ an MNIST subset of hard-to-distinguish digits "3" and "8" for binary classification. The information on this subset is summarized in Table 1. For comparison, we also apply 3 embedding methods, DFS,

Table 1: Information on benchmark and real-world datasets used in our experiments.

| Data Set | MNIST | glass | vowel | TOX-171 | yale | Enhancer–Promoter |
|---|---|---|---|---|---|---|
| #Features | 784 | 10 | 10 | 5784 | 1024 ($32 \times 32$) | 102 |
| #Classes | 2 | 6 | 11 | 4 | 15 | 3 |
| #Training | 11,982 | 150 | 742 | 137 | 132 | 5,756 |
| #Testing | 1,984 | 64 | 248 | 34 | 33 | 2,878 |

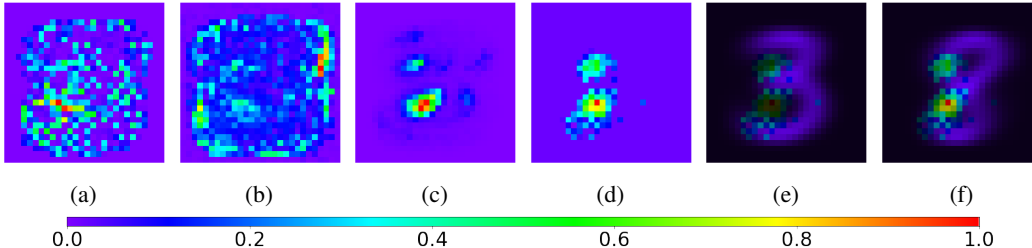

(a)  (b)  (c)  (d)  (e)  (f)

0.0   0.2   0.4   0.6   0.8   1.0

Figure 3: Feature importance maps yielded by different FIR methods. (a) DFS. (b) AvGrad. (c) RF. (d-f) Ours and our map superimposed on the mean images of "3" and "8", respectively, for clarity.

AvGrad and RF to this subset. To see the explainablity of FIR, we adopt the same full-connected MLP instead of CNN in DFS, AvGrad and the operator net in ours ($s = 85, d = 784$). The setting ensures that no other mechanisms like convolution/pooling layers can help a model automatically extract salient features for FIR. As a result, the accuracies yielded by DFS, AvGrad, RF and **ours** on the test data are $97.42 \pm 0.30\%$, $99.27 \pm 0.04\%$, $98.84 \pm 0.03\%$ and $\mathbf{99.31 \pm 0.08}\%$, respectively, where ours and DFS use 85 and 212 features, respectively, but AvGrad and RF need all 784 features. For visual inspection, we normalize the FIR scores achieved by different methods to the same range and illustrate typical feature importance maps produced by 4 methods in a fold in Fig. 3. It is observed from Fig. 3(a),(b) that DFS and AvGrad, two FIR methods for deep learning, do not produce explainable maps. In contrast, it is evident from Fig. 3(d-f) that ours yields a meaningful map where those features (pixels) that distinguish between "3" and "8" images are vividly highlighted in terms of their importance. Again, ours yields a map similar to that of RF (c.f. Fig. 3(c)) but outperforms this off-the-shelf FIR method.

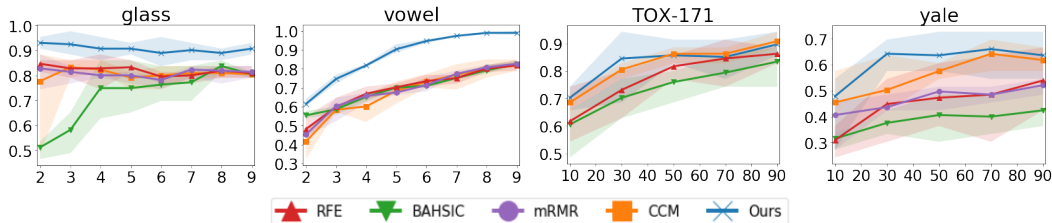

Figure 4: Classification accuracies (vertical axis) yielded by the supervised feature selection methods and ours for different numbers of selected features (horizontal axis) on 4 benchmark datasets.

**Feature Selection Benchmark**. We further conduct the evaluation in feature selection. As our approach has the same setting as used in the supervised feature selection methods, we compare ours to those strong supervised feature selection methods, RFE, BAHSIC, mRMR and CCM, on four benchmark datasets: glass [22], vowel [22], TOX-171 [23] and yale [24], as summarized in Table 1. For our model, we employ MLPs to implement the operator for glass, vowel and TOX-171 but a CNN to carry out the operator for yale to demonstrate the flexibility of our dual-net architecture. By following the setting used in [17], we employ kernel SVMs for classification on features selected by 4 filtering methods. It is evident from Fig. 4 that ours substantially outperforms all others on glass, vowel and yale with a large margin. Overall, ours yields results comparable to the strongest performer, CCM, on TOX-171 where there are 5,700+ features but only 109 training examples for parameter estimate in each of 5 folds, which is very challenging for deep learning.

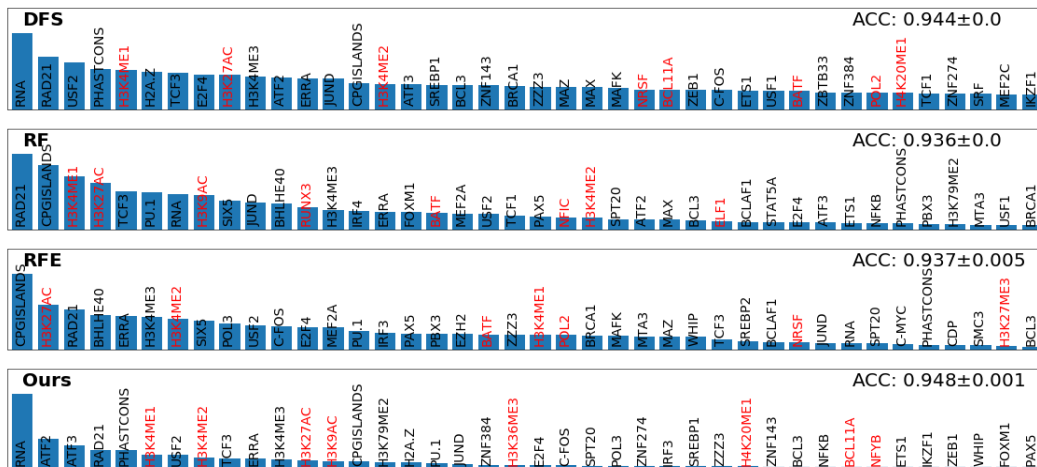

Figure 5: Accuracy and FIR scores yielded by different methods on GM12878 Cell line (200 bp).

## 4.3 Real-world Data

We finally evaluate our approach on a real-world enhancer–promoter data, a challenging task that classifies the function of DNA sequences into enhancer, promoter and background [25]. As listed in Table 1, the data used in this experiment are sampled from annotated DNA regions of GM12878 cell line (200 dp), the same as used in DFS [7], a feature selection method dedicated to this task. For comparison, we also apply DFS as well as RF and RFE, 2 strongest FIR methods manifested in our experiments, to this dataset. The same MLP architecture is used in implementing DFS and the operator net in our model. Fig. 5 shows accuracies and the FIR scores of top $s = 35$ out of $d = 102$ features yielded by 4 different methods and those features colored in red correspond to the genes of which functions are well known in medicine and genetics literature (see Table 2 in [7] for details). In terms of accuracy, ours is comparable to DFS and slightly better than RF and RFE, where the test accuracies of RFE and ours are based on 35 features but the accuracies of RF and DFS are achieved with all 102 and 94 features selected by DFS via weight shrinkage, respectively. As seen in Fig. 5, those top features ranked by DFS and ours, two deep learning methods, appear quite similar but significantly different from those top features ranked by RF and RFE. The biological implication resulting from the results shown in Fig. 5 is worth investigating further from a biological/medical perspective. More results on this dataset can be found from Sect. C in Supplementary Materials.

Regarding our alternate learning algorithm, our empirical studies suggest that it generally converges by reaching a local optimum (see Sects. B and C in Supplementary Materials for details).

## 5 Discussion

In general, our idea is motivated by RF [16] and the dropout regularization [26]; our exploration-exploitation strategy (c.f. Sect. 3.3) allows for the simultaneous use of different feature subsets and dropout of input "nodes" randomly during learning. As the joint use of multiple feature subsets in learning leads to more training examples of fewer features randomly, our approach could provide an alternative way to improve the generalization in deep learning when the limited training examples are available even though FIR/feature selection is not of interest in such application scenarios.

Also, we want to make a connection between our proposed approach and evolutionary computation in terms of feature selection [27]. In our approach, a single deep learning model, operator, works on different feature masks simultaneously during learning to carry out the functionality of a population of individual learning models in evolutionary computation. Instead of purely stochastic operations, mutation and crossover, on individual learners in a population used in evolutionary computation, our selector carried out by another single deep learning model uses a more efficient gradient-guided local stochastic search strategy to reduce the search space for combinatorial optimization. In general, our approach bears the spirit of evolution computation but addresses the combinatorial optimization issue

in an entirely distinct manner, which leads to a more effective yet efficient approach to populationwise FIR and feature selection.

Our proposed approach is scalable to big data and easily makes use of any state-of-the-art deep learning techniques to be our component models for populationwise FIR and feature selection. In terms of computational complexity, however, our approach suffers from a high computational burden in training due to use of the dual-net architecture involving two deep learning models and the alternate learning procedure (see Sect. C in Supplementary Materials for details). Nevertheless, the computational load issue in our approach could be addressed (at least alleviated) by the latest development in deep learning, e.g., EfficientNet [28].

Our approach can be applied to the generic populationwise feature selection problem that needs to find out an optimal feature subset from $\sum_{s=1}^{d-1} \binom{d}{s}$ subsets for a feature set of $d$ features. Instead of a direct search of the entire subset space, we adopt a strategy that makes our model work in parallel on different subset sizes, the same as used in the state-of-the-art supervised feature selection methods, e.g., CCM [17]. To this end, however, our approach might have a higher computational burden than those kernel-based methods in learning. Also, our approach is extensible to group-based FIR and feature selection by introducing the group feature constraints to our stochastic local search procedure (c.f. Sect. 3.3), which would overcome the limitation of linear models, e.g., group LARS/LASSO [29], in capturing the complex functional dependency between group input features and targets. Furthermore, our proposed dual-net architecture can also be extended to unsupervised feature selection by carrying out the operator with an autoencoder-like learning model.

In conclusion, we propose a dual-net neural architecture along with an alternate learning algorithm to enable deep learning to work effectively for FIR and feature selection. A thorough evaluation manifests that our approach outperforms several state-of-the-art FIR and supervised feature selection methods. In our ongoing work, we would extend our approach to instancewise FIR, group and unsupervised feature selection scenarios and explore its potential in challenging real applications.

## Broader Impact

This research makes contributions to machine learning models and algorithms in general and does not involve any issues directly regarding ethical aspects and future societal consequences. In the future, our approach presented in this paper might be applied in different domains, e.g., medicine and life science, where ethical aspects and societal consequences might have to be considered.

## Funding Disclosure

This research was fully conducted at The University of Manchester. There is neither external funding in support of this work nor competing interests with any third parties.

## Acknowledgement

The authors are grateful to three anonymous reviewers for their comments. In particular, the authors would thank the anonymous reviewer who offers us an insight by understanding our contributions from an evolutionary computation perspective.

## Footnotes

[1]Parameters are actually updated on a batch $\mathcal{B}$ randomly drawn from $\mathcal{D}$, hence $\frac{|\mathcal{D}|}{|\mathcal{B}|}$ times in an epoch.

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
