[Supplementary Material]

# Supplementary Materials of Feature Importance Ranking for Deep Learning

**Maksymilian A. Wojtas**      **Ke Chen**
Department of Computer Science, The University of Manchester, Manchester M13 9PL, U.K.
{maksymilian.wojtas,ke.chen}@manchester.ac.uk

In this document, we present our experimental setup, more experimental results and the implementation of our alternate learning algorithm. Section A describes the the detailed experimental setup for all the method including ours in our comparative study. Sections B and C report the typical learning behavior of our alternative learning algorithm and more experimental results . Section D presents the pseudo code of our alternate learning algorithm.

## A   Experimental Setup

In this section, we describe the details of experimental settings used in our experiments. In our experiments, we always use the *grid search* along with *5-fold cross-validation* on a training set to find out optimal hyperparameters involved in different learning methods. Below, we first present the detailed setup in our approach, then describe all the technical details of other learning methods used in our comparative studies on different datasets.

### A.1   Setup in Our Approach

In our implementation of the operator net, we have to consider an issue concerning the differentiation between *a selected feature of which value is zero* and *any removed features masked with zero* due to the use of the binary masks in our work. Thus, we design an operator net architecture shown in Figure I. Instead of feeding only the selected features, $x \otimes m$, to the first hidden layer, we concatenate the mask, $m$, used to indicate the selected features, and the selected features themselves, $x \otimes m$, to form the input fed to the first hidden layer as illustrated in Figure I. Thus, the dimension of the input to the first hidden layer is $2d$ rather than $d$ features described in the main text. It is worth mentioning that we had investigated other manners to tackle the aforementioned "zero-value" issue, e.g., stipulating a value beyond the range of any features for a removed feature in $x \otimes m$. However, neither of those yields the better performance than the architecture presented above.

Moreover, there are specific settings in our approach due to technical reasons; e.g., the loss function used to train the selector net and the subtle technical details related to Phase II in our alternate learning algorithm, as described in Sect. 3.3 of the main text.

In our experiments, the loss function used to train the selector net presented in Eq.(2b) of the main text is actually replaced by a *weighted* loss as follows:

$$\mathcal{L}_S(\mathcal{M}'; \varphi) = \frac{1}{2|\mathcal{M}'|} \sum_{m \in \mathcal{M}'} w_m \Big( f_S(\varphi; m) - \frac{1}{|\mathcal{D}|} \sum_{(x,y) \in \mathcal{D}} l(x \otimes m, y; \theta) \Big)^2,$$

where $w_m = 10, 5, 1$ when $m = m_{t,best}$ (the best performed subset found in the last step), $m = m_{t+1,opt}$ (the optimal subset generated in the current step), and $m$ is any of other subsets in $\mathcal{M}'_{t+1}$, respectively (c.f. Phase II-A in Sect. 3.3 of the main text). The above weighted selector loss exploits what has been learned so far in order to facilitate the stochastic local search in tackling the combinatorial optimization problem more effectively.

Figure I: The actual implementation of the operator net in our experiments to overcome the "zero-value" representation issue. As a result, both the selected features, $\boldsymbol{x} \otimes \boldsymbol{m}$, and the mask, $\boldsymbol{m}$, used to indicate those selected features are concatenated as the input to the first hidden layer.

Table I: The optimal architectural hyperparameters of our dual-net learning model in our experiments. **CNN**$^*$: there are 3 convolutional layers of 32, 64 and 128 channels and the corresponding kernel sizes are 5, 3, 3, respectively. Each of the convolutional layers is followed by a maximum pooling layer. Two input channels are used for coping with the "zero-value" issue, one for the selected features, $\boldsymbol{x} \otimes \boldsymbol{m}$, and the other for the mask, $\boldsymbol{m}$. (c.f. Figure I) Two dense layers of 30 neurons are top on the last convolutional layer. The output layer of 15 softmax neurons corresponds to 15 classes.

| Data Set | Operator Net | Selector Net |
|:---:|:---:|:---:|
| 4-way Classification | $20 \to 60 \to 30 \to 20 \to 4$ | $10 \to 100 \to 50 \to 10 \to 1$ |
| Nonlinear Regression | $20 \to 100 \to 50 \to 25 \to 1$ | $10 \to 100 \to 50 \to 10 \to 1$ |
| Binary Classification | $20 \to 60 \to 30 \to 20 \to 1$ | $10 \to 100 \to 50 \to 10 \to 1$ |
| MNIST Subset | $1568 \to 500 \to 250 \to 100 \to 1$ | $784 \to 300 \to 200 \to 100 \to 1$ |
| Glass | $20 \to 50 \to 25 \to 10 \to 6$ | $10 \to 500 \to 250 \to 100 \to 1$ |
| Vowel | $20 \to 50 \to 25 \to 10 \to 11$ | $10 \to 500 \to 250 \to 100 \to 1$ |
| TOX-171 | $11568 \to 100 \to 50 \to 20 \to 4$ | $5784 \to 500 \to 250 \to 100 \to 1$ |
| Yale | **CNN**$^*$ | $1024 \to 500 \to 250 \to 100 \to 1$ |
| Enhancer–Promoter | $204 \to 300 \to 200 \to 50 \to 3$ | $102 \to 500 \to 250 \to 100 \to 1$ |
| RNA-seq | $40528 \to 1000 \to 500 \to 200 \to 5$ | $20264 \to 500 \to 250 \to 100 \to 1$ |

In our experiments, the 3-step validation procedure used for *generation of the optimal subset* would ensure its optimality within those feature subset candidates (c.f. Phrase II.A in Sect. 3.3 of the main text). However, the condition to exit from the loop of repeating steps i)-iii) may not be always satisfied. In our experiments, we hence set the maximum number of repetition in this test to *5* iterations so that the subset optimality validation procedure always ends up to *five* iterations. In addition, the parameter update of the operator and the selector nets in Phase II is done in different frequencies in the alternate learning; i.e., the parameters in the operator net are updated once on each batch in Phase II.B, while the parameters in the selector net are updated once on every 8 batches in Phase II.A.

We employ MLPs (CNNs) of the sigmoid (ReLu) neurons to carry out the operator net and MLPs with the sigmoid neurons for the selector net in our dual-net architecture. For training MLPs (CNNs), we adopt the *Adam optimizer* (Adam with Nestrov momentum for the operator net)[1] via the stochastic gradient descent (SGD) procedure. Early stopping is used based on the losses evaluated on the validation data[1]. All the optimal hyperparameters used in our experiments are summarized in Tables I and II, respectively.

### A.2 Optimal Hyperparameters in Other Methods

Other 9 methods have also been employed for a comparative study on different datasets. We strictly follow their original settings described in those papers. We implement deep learning algorithms with by ourselves with Tensorflow 2.0 [2] and Keras [3]. For other methods, we use the existing code

Table II: Other optimal hyperparameters of our dual-net learning model in our experiments. $E_1$ is the number of SGD training batches (instead of epochs) in Phase I of the alternative learning. In Phase II.A, $|\mathcal{M}'_t|$ refers to the number of different optimal subset candidates used in a single batch during the SGD learning. $f = \frac{|\mathcal{M}'_{t,1}|}{|\mathcal{M}'_{t,2}|}$ is the fraction that governs the exploitation-exploration trade-off in the selector learning, and $s_p$ indicates the number of elements perturbed. For details of the alternative learning algorithm, see Section 3.3 in the main text.

| Data Set | $E_1$ | $|\mathcal{M}'_t|$ | $f$ | $s_p$ |
|---|---|---|---|---|
| 4-way Classification | $6,000$ | 32 | 0.5 | 2 |
| Nonlinear Regression | $6,000$ | 32 | 0.5 | 2 |
| Binary Classification | $6,000$ | 32 | 0.5 | 2 |
| MNIST Subset | $10,000$ | 32 | 0.5 | 5 |
| Glass | $10,000$ | 128 | 0.5 | 2 |
| Vowel | $6,000$ | 128 | 0.5 | 2 |
| TOX-171 | $1,500$ | 128 | 0.5 | 5 |
| Yale | $4,500$ | 128 | 0.5 | 5 |
| Enhancer–Promoter | $10,000$ | 64 | 0.5 | 5 |
| RNA-seq | $8000$ | 32 | 0.5 | 5 |

Table III: Optimal regularization hyperparameters, $\lambda$, used in LASSO on different datasets in our experiments.

| Data Set | Fold-1 | Fold-2 | Fold-3 | Fold-4 | Fold-5 |
|---|---|---|---|---|---|
| 4-way classification | 0.055 | 0.056 | 0.046 | 0.008 | 0.042 |
| Nonlinear Regression | 0.001 | 0.0 | 0.053 | 0.001 | 0.0 |
| Binary Classification | 0.025 | 0.07 | 0.081 | 0.02 | 0.039 |

in the Python scikit-Learn library [4] for FS, LASSO, RF, RFE or the authors' project website for BAHSIC[2], mRMR[3] and CCM [4]. Our source code will be made available after the completion of review. Below, we summarize the actual optimal hyperparameters pertaining to those methods used in our comparative study.

**Deep Feature Selection (DFS)** [5]. For DFS, we use the MLPs of the architectures as same as that of the operator net in our dual-net architecture apart from the input layers for a given task, as shown in Table I. Instead of having the concatenation of the selected features and the mask indicating the selected features in our operator net, the DFS appends an additional one-to-one layer between the input and the first hidden layer. Similarly, the sigmoid neurons are used in their modified MLPs and the Adam optimizer [1] is adopted for training MLPs via the SGD. The optimal regularization hyperparameter is $\lambda = 0.01$ for 3 synthetic datasets and the MNIST subset after a grid search from a large range of $\lambda$. For the Enhance-Promoter dataset, the optimal hyperparameter is $\lambda = 0.008$. The rest of the parameters are kept the same as suggested in [5], which is $\lambda_2 = 1, \alpha_1 = 0.0001, \alpha_2 = 0$. Note that we implement the DFS code by ourselves with Tensorflow 2.0 and Keras since the authors' code is merely applicable to a specific dataset.

**Average Input Gradient (AvGrad)** [6]. It is simply a post-processing method for feature importance ranking (FIR) based on a trained MLP, we employ the same MLP architecture as that of our operator net apart from the input and the Adam optimizer [1] via the SGD on 3 synthetic datasets and the MNIST subset, Table I.

**Forward Selection (FS)** [7]. For FS, we employ the MLPs as the base leaner in this wrapper method and the training procedure identical to those used in the AvGrad on 3 synthetic datasets. For

Table IV: Optimal hyperparameters (#tree, depth) for RF on different datasets in our experiments.

| Data Set | Fold-1 | Fold-2 | Fold-3 | Fold-4 | Fold-5 |
|---|---|---|---|---|---|
| 4-way classification | (80, 14) | (70, 11) | (90, 15) | (150, 14) | (150, 12) |
| Nonlinear Regression | (50, 15) | (60, 15) | (50, 14) | (60, 10) | (50, 8) |
| Binary Classification | (60, 14) | (50, 10) | (90, 8) | (90, 15) | (160, 13) |
| MNIST | (200, 21) | (200, 20) | (210, 21) | (200, 21) | (200, 20) |
| Enhancer-Promoter | (120, 11) | (120, 12) | (100, 15) | (150, 13) | (60, 14) |

FIR, FS always ranks the importance of an early selected feature higher than that of others selected later in the forward subset selection procedure.

**LASSO** [8]. We use the grid-search to find out the optimal regularization hyperparameter, $\lambda$, in LASSO. The optimal hyperparameters found in 5 folds are listed in Table III.

**Random Forest (RF)** [9]. We use the grid-search to find out the optimal hyperparameters: number of decision trees and depth of the trees. We search from a range from 50 to 220 trees and between 7 and 24 in depth. The optimal hyperparameters found in 5 folds are listed in Table IV.

**Recursive Feature Estimation (RFE)** [10]. We use 1 step for all the datasets apart from TOX-171 and Yale datasets where 5 steps are used. In our experiments, we adopt the default values for underlying estimators (linear SVM) with $C = 1$ and $\gamma = \frac{1}{n_{features} * \text{var}(X)}$.

**Backward Elimination using HSIC (BAHSIC)** [11, 12]. A default hyperparemeter regarding the fraction of removed features in each iteration is set to 0.1 as suggested in their papers. In our experiments, we adopt the inverse kernels suggested in their papers and the BAHSIC webpage.

**Minimal Redundancy Maximal Relevance Criterion (mRMR)** [13]. No hyperparemeter needs to be tuned in this method. In our experiments, we adopt he "MIQ" option suggested in the `PyMRMR` library.

**Conditional Covariance Minimization (CCM)** [14]. We use $\epsilon = 0.001$ for two synthetic classification datasets, 4-way and binary classification, and 4 benchmark datasets, Glass, Vowel, TOX-171 and Yale. For the nonlinear regression dataset, we use $\epsilon = 0.1$. As all 7 datasets were used in the paper, we adopt the optimal hyperparameters reported in the paper and suggested in the CCM repository.

As CCM, RFE, BAHSIC, mRMR and LASSO are filtering methods for feature selection, we need to measure their performance based on another learning model. For CCM, RFE, BAHSIC and mRMR, we adopt the same setting used in [14], i.e., SVM/SVR with a Gaussian kernel of optimal hyperparameters: $C = 1$ and $\gamma = \frac{1}{n_{features} * \text{var}(X)}$. For LASSO, we use the same MLPs used in deep learning models, i.e., DFS, AvGrad and ours (operator net).

## B More Results on Synesthetic and Benchmark Data

In this section, we demonstrate the typical learning behavior of our alternate learning algorithm on different datasets, describe the detailed information of benchmark datasets used in our experiments and report more experimental results.

### B.1 Learning Behavior

As described in Section 3.3 of the main text, our alternate learning algorithm trains two learning models, operator and selector, simultaneously in an alternate manner; i.e., in Phase II, the learning behavior of the operator and the selector nets are mutually affected each other in each batch during the SGD learning. This is different from most of the existing deep learning algorithm that involves only a deep neural network to be trained. Therefore, we need to investigate how our proposed learning

(a) selector-loss

(b) operator-loss-train

(c) operator-loss-train($\boldsymbol{m}_{opt}$)

(d) operator-loss-val

(e) operator-loss-val($\boldsymbol{m}_{opt}$)

Figure II: **Synthetic nonlinear regression dataset**. Evolution of the operator and the selector losses in Phase II ($d = 10, s = 5$). The x-axis corresponds to the number of batches and y-axis refers to the loss statistics of 5 folds. (a) The selector loss. (b) The operator loss on the training set. (c) The operator loss on the training set with $\boldsymbol{m}_{opt}$ only. (d) The operator loss on the validation set. (e) The operator loss on the validation set with $\boldsymbol{m}_{opt}$ only. Note that Phase II starts when the operator net has been trained for 6,000 batches in Phase I.

model behaves. Below, we exhibit the typical learning behavior in our alternative learning on 3 datasets, *synthetic nonlinear regression*, *MNIST subset* and *Yale*.

Figure II illustrates the learning behavior of the operator and the selector in terms of losses in Phase II on the synthetic *nonlinear regression* dataset. It is observed from Figure II(a) that the trained operator in Phase I provides informative training examples so that the averaged selector loss of 5 folds decreases monotonically as required. As evident in Figures II(b) and II(d), the averaged operator loss on training and validation sets further decreases steadily as the selector keeps offering more "promising" optimal mask candidates achieved by the stochastic local search for combinatorial optimization. It is clearly seen in Figures II(b) and II(d) that at the beginning of Phase II (up to 1k batches), operator loss on both training and validation sets sharply decreases once the selector has been involved. Also, the loss may be reduced substantially when an optimal mask is identified, as shown in Figure II(d) (between 6k and 7k batches). Given the fact that at the end of Phase II.A for each iteration, we always achieve an optimal mask, $\boldsymbol{m}_{t,opt}$. Thus, we can apply such optimal masks only to measure the operator loss. As a result, Figures II(c) and II(e) illustrate the evolution of the operator loss evaluated with $\boldsymbol{m}_{t,opt}$ only on training and validation sets. In contrast to the operator loss with all optimal mask (subset) candidates shown in Figure II(d), the abrupt loss drop resulting from the identified optimal mask is much more visible in Figure II(e). Therefore, early stopping in our alternate learning algorithm is based on the operator loss evaluated with $\boldsymbol{m}_{t,opt}$ only. Overall, Figure II demonstrates that our alternate learning algorithm works well and eventually converges for this regression task.

Next, Figure III illustrates the learning behavior of the operator and the selector in terms of losses in Phase II on the MNIST benchmark subset, a *binary classification* task. It is seen from Figure III(a) that the evaluation of the averaged selector loss of 5 folds has a reduction trend as the number of

(a) selector-loss

(b) operator-loss-train

(c) operator-loss-train($\boldsymbol{m}_{opt}$)

(d) operator-ACC-train

(e) operator-loss-val

(f) operator-loss-val($\boldsymbol{m}_{opt}$)

(g) operator-ACC-val

Figure III: **MNIST Benchmark Subset**. Evolution of the operator and the selector losses in Phase II ($d = 784, s = 85$). The `x-axis` corresponds to the number of batches and `y-axis` refers to the loss statistics of 5 folds. (a) The selector loss. (b) The operator loss on the training set. (c) The operator loss on the training set with $\boldsymbol{m}_{opt}$ only. (d) The classification accuracy evaluated on the training set. (e) The operator loss on the validation set. (f) The operator loss on the validation set with $\boldsymbol{m}_{opt}$ only. (g) The classification accuracy evaluated on the validation set. Note that Phase II starts when the operator net has been trained for 10,000 batches in Phase I. The spike at the maximum batch in (e)-(g) correspond to the results evaluated on the test set with the trained operator net upon the completion of the alternate learning.

batches is increased although the averaged loss not longer drops monotonically. The sharp selector loss increase at around 10k batches is typical and reflects the nature of our stochastic local search procedure in tackling the combinatorial optimization issue. The sharp increase is likely caused by the fact that the optimal mask identified leads to the sharp operator loss reduction and the selector net did not have such training examples before this moment. This analysis is manifested by all the results at round 10k batches shown in other plots in Figure III. As evident in Figures III(b) and III(c), the averaged operator loss on training and further decreases in general. By using an alternative performance index, we also show the averaged classification accuracy measured on the training set in Figure III(d), which allows one to see the learning performance vividly. Likewise, we illustrate the averaged operator loss and accuracy on the validation set in Figures III(e)-(g). Once again, we can see our alternate learning algorithm works very well. Once again, the operator validation loss evaluated with $\boldsymbol{m}_{opt}$ only provides the solid evidence for early stopping. In general, the learning behavior on this binary classification dataset very much resembles that on the nonlinear regression dataset (c.f. Figure II). After the alternate learning is completed, we can evaluate the performance of the trained operator net on the test set in the same manner. To show the test performance, we depict the averaged loss evaluated on the test set with all the optimal mask candidates and the optimal mask as well as the accuracy based on the optimal mask at the maximum batch in Figures III(e)-(g). Interestingly, it is seen in Figures III(e)-(g) that the test performance is significantly better than the validation performance in terms of both the losses and the accuracy. This suggests that our alternate learning algorithm yields the favorable generalization performance on this benchmark dataset.

(a) selector-loss

(b) operator-loss-train

(c) operator-loss-train($\boldsymbol{m}_{opt}$)

(d) operator-ACC-train

(e) operator-loss-val

(f) operator-loss-val($\boldsymbol{m}_{opt}$)

(g) operator-ACC-val

Figure IV: **Yale Benchmark Dataset**. Evolution of the operator and the selector losses in Phase II ($d = 784, s = 10, 30$). The `x-axis` corresponds to the number of batches and `y-axis` refers to the loss statistics of 5 folds. The light/dark colors correspond to $s = 10/s = 30$, respectively. (a) The selector loss. (b) The operator loss on the training set. (c) The operator loss on the training set with $\boldsymbol{m}_{opt}$ only. (d) The classification accuracy evaluated on the training set. (e) The operator loss on the validation set. (f) The operator loss on the validation set with $\boldsymbol{m}_{opt}$ only. (g) The classification accuracy evaluated on the validation set. Note that Phase II starts when the operator net has been trained for 4,500 batches in Phase I. The spike at the maximum batch (6k) in (e)-(g) correspond to the results evaluated on the test set with the trained operator net upon the completion of the alternate learning.

Finally, Figure IV shows the learning behavior of the operator and the selector in terms of losses in Phase II on the Yale benchmark dataset, a *multiclass classification* task. For this facial image dataset, we employ a convolutional neural network described in Table I to carry out the operator net. To understand the learning behavior better, we compare the situations of the alternate learning for different subset sizes, $s = 10$ and $s = 30$. It is observed from Figure IV(a) that the averaged selector loss for different subset sizes behaves quite differently. For $s = 10$, the selector loss sharply decreases at the first few hundred batches then sharply increases. The limited amount of information carried in 10 out of 1024 features may be accountable for this phenomenon. In contrast, the evolution of selector loss for $s = 30$ is similar to that shown in Figures II(a) and III(a). Figures IV(b)-(d) suggest that the averaged operator loss for different subset sizes keeps decreasing and the accuracy remains increasing on the training set. In contrast, the trend of the averaged operator loss for different subset sizes increases on the validation set after 1.5k batches as shown in Figures IV(e) and (f). This looks like a typical overfitting scenario. As seen in Figure IV(g), however, the averaged classification accuracy on the validation set generally keeps increasing regardless of different subset sizes. Furthermore, for $s = 30$, the averaged operator test losses and the test accuracy shown in Figures IV(e)-(g) (at 6k batches) also provide the evidence for the good generalization performance. Surprisingly, the alternate learning behavior on this benchmark dataset contradicts or is inconsistent with the normal behavior of a learning system. While we do not fully understand such learning behavior, our preliminary analysis implies that the this phenomenon could be caused by the *covariant shift* nature of this facial image dataset and limited training data. In the Yale dataset, the images of an individual subject correspond

to different facial expressions. Since there are only limited training examples and the selector learning is constrained by the operator training performance, the stochastic local search in Phase II.A may have to do a lot of exploration in order to find out the "genuine" optimal subset (mask). This can be observed by the fluctuated operator validation loss as shown in Figures IV(e) and (f). Thanks to our stochastic exploration-exploitation strategy, some sub-optimal subsets may still direct the learning towards the learning performance at an acceptance level.

In summary, we exhibit typical yet different learning behavior of our dual-net architecture trained by the alternate learning algorithm in Figures II-IV. In most of the situations including the one reported in Section C and others not reported here, we can use the operator validation loss evaluated with the optimal mask only for early stopping. In some occasion, however, we encounter some "strange" learning behavior, as exemplified in Figure IV. In such occasion, we might have to use the validation classification accuracy (or validation MSE in regression) for early stopping. Thus, we are going to investigate such "strange" learning behavior in our ongoing work.

## B.2 Detailed Information on Benchmark Data

In this section, we provide the detailed information on 4 benchmark datasets described in the main text. In our comparative study, we choose 4 challenging benchmark datasets for feature selection evaluation. As reported in [14], the state-of-the-art feature selection methods including those latest strong ones do not perform well on the following datasets.

**Glass dataset**[5] The Glass is a famous UCI benchmark dataset for a task of predicting a type of glass based on its chemical composition. Glass dataset contains usually 9 chemical features and the ID for each instance that is normally not treated as a feature. In the experimental setting of CCM [14], they treated the ID as a new feature so that 10 features are used in their experiments. Due to the fact that the instances in the data file are arranged in an non-shuffled manner according to their class labels, the ID feature turns up to be one of the most important features so that CCM and other strong feature selection methods yields very high accuracy, e.g., CCM achieves 86% on average [14]. In our experiment, we follow this setting so that our approach yields 90%+ accuracy (c.f. Figure 4 in the main text). Without the ID feature, however, all the methods including ours yield considerably lower accuracies although ours still outperforms those methods used in our comparative study. In the 5-fold cross validation, the accuracy of our approach drops to the levels of 75%-80%, quite close to the known top accuracy of 80% on the OpenML platform[6].

**Vowel Dataset**[7] The Vowel is yet another famous UCI benchmark dataset for predicting English vowels from acoustic features. Following the same setting used in CCM [14], we use a newer version of this dataset so that we can make a fair comparison to those feature selection methods used in our comparative study.

**TOX-171 dataset**[8] The TOX-171 is a biological microarray dataset with only 43 instances/class but 5,784 features. The nature of this dataset makes a deep learning model very prone to overfitting. As shown in Figure 4 in the main text, our approach does not outperform the CCM in general, which reveals the limitation of our approach.

**Yale Dataset**[9] The Yale is a well-known facial image benchmark dataset. There are 15 individual subjects and 11 images of different facial expressions, e.g., wink, happy and sad, were collected from each individual. When this dataset is used for face recognition, a random split of this dataset could lead to a certain degree of covariant shift; the instances in training and validation/test sets may be subject to different distribution but their distributions conditional on the label are same. This causes a difficulty for all learning models without covariant shift adaptation.

Figure V: **MNIST Subset**. Feature importance maps ($d = 784, s = 85$) generated with the method described in Section B.3. From top to bottom, first 4 rows correspond to feature importance maps achieved from different folds. The bottom row is the full feature importance map corresponding the feature importance map shown in Figure 4 of the main text.

### B.3 Feature Importance Map

Due to the limited space in the main text, we demonstrate only one feature importance map on the MNIST subset. Below, we show more feature importance maps achieved from other folds on the MNIST subset and those yielded by our approach on the Yale dataset.

To obtain the superimposed feature importance maps on the background image (the mean of raw images), we apply a method as follows. A blank image is first created in the HSV (hue-saturation-value) colour format. The hue used in [0,270] range corresponds to the importance, and the saturation is set to 1.0 to encode the mean background image from the dataset. Due to the feature importance ranking (FIR) scores are normalised, no negative FIR scores are shown to ensure unselected features have the background color.

Figure V shows different feature importance maps achieved from other 4 folds. As our FIR approach described in Section 3.4 of the main text measure the FIR scores based on the input gradient, it can achieve the input gradient for all the features regardless of whether a feature is selected or not. To this end, we can generate a full feature importance map as well. It is observed from V that the feature importance maps achieved from different folds are very much consistent and the full feature importance map provides a clearer picture in terms of explainablity/interpretability.

Figure VI: **Yale Dataset**. Feature importance maps achieved with different subset mask sizes: $s = 10, 30, 50, 70, 90$ (from left to right) out of $d = 32 \times 32$. The second row corresponds to the images generated by superimposing the feature importance maps to the background image, i.e., mean face image averaged on 11 images collected from an individual.

As Yale is a facial image dataset, we can also illustrate the feature importance maps in Figure VI for visual inspection. The visual inspection reveals that increasing the mask size $s$ results in less clear visual representation of feature importance. In comparison, the best performed mask size of 30 clearly selects several meaningful yet discriminative features, e.g., pixels near lip, nose and eyes, and ranks their importance properly as shown in the 2nd column in Figure VI. As this dataset has limited instances (7 training examples/class on average), we reckon that the use of large subset mask size is likely to cause the overfitting as revealed by their feature importance maps shown in Figure VI.

## C   More Results on Real-World Data

In this section, we report more results on the enhance-promoter dataset and the UCI gene expression cancer RNA-Seq data set.

### C.1   Results on Enhance-Promoter Dataset

To evaluate our approach on real-world data, we adopt the GM12878 cell line (a lymphoblastoid cell line) dataset [5]. This is the dataset for which the *deep feature selection* (DFS) method [5] is especially proposed. Therefore, we follow their setting by using only annotated DNA regions of GM12878 cell line (200 bp).

(a) selector-loss

(b) operator-loss-train

(c) operator-loss-train($\boldsymbol{m}_{opt}$)

(d) operator-ACC-train

(e) operator-loss-val

(f) operator-loss-val($\boldsymbol{m}_{opt}$)

(g) operator-ACC-val

Figure VII: **Enhancer-Promoter Dataset**. Evolution of the operator and the selector losses in Phase II ($d = 102$, $s = 35$). The `x-axis` corresponds to the number of batches and `y-axis` refers to the loss statistics of 5 folds. (a) The selector loss. (b) The operator loss on the training set. (c) The operator loss on the training set with $\boldsymbol{m}_{opt}$ only. (d) The classification accuracy evaluated on the training set. (e) The operator loss on the validation set. (f) The operator loss on the validation set with $\boldsymbol{m}_{opt}$ only. (g) The classification accuracy evaluated on the validation set. Note that Phase II starts when the operator net has been trained for 10,000 batches in Phase I. The spike at the maximum batch in (e)-(g) correspond to the results evaluated on the test set with the trained operator net upon the completion of the alternate learning.

In the original dataset, there are 7 classes and 102 features, each class has 3,000 instances apart from one that has only 2878 instances. The 7 classes are ***active promoter, active enhancer***, *active exon, inactive promoter, inactive enhancer, inactive exon* and *unknown regions*. The main interest in medicine is classifying the function of DNA sequences into enhancer, promoter and background since non-coding gene regulatory enhancers are essential to transcription in mammalian cells [15].

Following the suggestion in [15, 5], we merge inactive enhancers, inactive promoters, active exons, and unknown regions into a background class. Thus, we have a 3-class imbalanced dataset as the background class has roughly 5 times more instances than other two classes: active promoter and active enhancer. We follow a preprocessing method consisting of two steps: 1) making the dataset balanced by down-sampling so that each of 3 classes has 2878 samples; 2) overcoming the natural skewness of biological outcome by taking the logarithm on the input. To avoid the zero-value issue in logarithm, we append a small positive number to each feature, $x \leftarrow x + 0.01$, in our experiments. Note that step 2) is not described in the DFS paper but we believe that this is important for such a data distribution.

It is also worth clarifying that we see some discrepancies between the data presented in the authors' repository[10] and their article [5]. Two main differences include 1) 102 features in the repository but 92 features stated in the article; 2) at least 2,878 instance/class in the repository but only 2,156

Figure VIII: Classification accuracies yielded by different methods on the Enhancer-Promoter dataset: cell line GM12878 (200dp). The shadowed regions refer to the performance range between the minimum and the maximum accuracies on 5 folds. While DFS and RF yield only one result with all the 102 features, other methods produce the results at different subset sizes for $s = 15, 25, 35, 45, 55$.

features mentioned in the article. While we use the dataset in their repository, we have done our best by keeping all the settings suggested in their article for a fair comparison in our experiments.

Due to the limited space in the main text, we only report the result of our approach for a subset mask size, $s = 35$. Here, we report more results of our approach and other methods used in comparative study on this dataset.

We first illustrate the learning behavior of our dual-net model on this real-world dataset in Figure VII. As shown in Figure VII(a), the averaged selector loss has the typical behavior as described in Section B.1. The loss fluctuation and the loss reduction trend in the loss evolution vividly exhibit how the stochastic local search strategy works in finding out optimal subset masks. As shown in Figures VII(b)-VII(g), the learning progresses steadily as evident in the evolution of the averaged operator losses and the averaged classification accuracies on the training and the validation sets. Also it is clearly seen in Figures VII(b)-VII(g), the overfitting occurs once the optimal subset mask is identified at around 10.5k batches. Once again, this observation provides the solid evidence to support for early stopping with the operator training/validation losses measured on the optimal subset masks ($\boldsymbol{m}_{opt}$). Furthermore, it is also seen in Figures VII(e)-VII(g) (at the maximum batches) that the averaged test losses and the averaged classification accuracy yielded by the trained operator net on 5 folds are superior to their counterparts on the validation set. Once again, this suggests that our alternate learning leads to the favorable generalization performance although an earlier stopping may yield a better accuracy.

Apart from the comparative study specified in the main text, we have conducted the further experiments on this dataset for a comparison to two recent state-of-the-art methods [16, 17] that obtain the populationwise FIR by aggregating the instancewise FIR. In [16], the global aggregation method workable on this dataset is the homogeneity-weighted importance, which is the same as the global LIME importance proposed in [18]. In our experiment, we use an MLP of the architecture: 102-300-200-50-3 and `n_samples=500` to achieve the LIME importance on the validation set [18]. In the SAN [17], the populationwise FIR is obtained via either instance-level aggregation (SAN_AGGR) or global attention layer (SAN_GLOBAL). In our experiments, we use the same settings suggested by the authors [17][11] with the following hyperparameters: $k = 1$, `p_dropout=20%`, `epochs=32`, `batch_size=32`, `n_1_1=128` (number of hidden neurons in SAN). In terms of feature selection, both methods fall into the filtering category. Therefore, we employ an MLP of the architecture:

s-300-200-50-3 to be a classifier trained on those selected feature subsets of $s = 5, 25, 35, 45, 55$, respectively, for this 3-class classification task.

Figure IX: Feature importance ranking (FIR) scores yielded by LIME, SAN and ours for top 55 features ($s = 55$) on the Enhancer-Promoter dataset: GM12878 Cell line (200 bp).

We report the accuracies yielded by 6 different methods for different subset sizes. As shown in Figure VIII, it is evident that our approach yields slightly better accuracies than the DFS [5], a method especially proposed for this biological dataset, when the subset size of selected features is larger than 15. Also, our approach outperforms RFE [10], a state-of-the-art feature selection method especially effective on gene selection for cancer classification, and RF [9], a famous off-the-shelf ensemble learning model. In contrast, it is evident from Figure VIII that ours along with DFS also outperforms those methods of using the aggregation of instancewise FIR to obtain populationwise one at all different subset sizes ranging from 15 to 55.

For feature importance ranking (FIR), we show the FIR scores yielded by two instancewise aggregation-based methods and ours for $s = 55$ in Figure IX. It is observed that two methods and ours yield different FIR results and different settings in the SAN also results in different FIR for top 55 features. Due to a lack of the ground-truth, it is difficult to draw an affirmative conclusion but the experimental results suggest that the populationwise FIE is an extremely challenging problem for real-word data.

We further show the FIR scores for top-40 features produced by DFS, RF, RFE and ours in Figure X. The FIR scores of the RFE and the RF are generated based on the RFE feature importance estimator [10] and the out-of-bag errors [9]. The FIR score of the DFS is achieved based the magnitude of shrunk weights between input and the first hidden layer introduced in the DFS method [5]. From Figure X, it is observed that our approach yields the relatively consistent FIR results when different subset sizes are used given the fact that the importance ranking order of top features only varies for one or two. Also, our approach is the only one to constantly rank "RNA" and two important genes "ATF2"" and "ATF3" among the most important features regardless of feature subset sizes. In comparison, the DFS also selects those two genes but does not rank them on the top. On the other hand, the RFE chooses other genes, "RAD21", ""PGISLANDS" and "H3K4ME3", as the most important features irrespective of feature subset sizes. It is also be seen in Figure X that the DFS and ours, two deep learning models rank the importance of features similarly but differently from the RFE and the RF that yield similar FIR scores. Our experimental results on this real-world dataset suggest that deep learning models may lead to different results from the existing state-of-the-art and off-the-shelf machine learning models for FIR. Thus, learning models of different types should be considered simultaneously and their results can be fused at the discretion of domain experts in such real-world applications.

To evaluate the efficiency, we record the averaging training time on this dataset in terms of 5-fold cross-validation. our experiments are conducted on a server of the specification and the environment: Intel Core i7-5930K CPU (3.50GHz), NVIDIA GeForce GTX TITAN X GPU, 64 GB RAM and CentOS 7. In summary, our algorithm takes around 1,100 sec while RF, SAN, DFS, LIME and RFE

take around 2.5, 35, 90, 540 and 1,700 sec, respectively. The dual-net architecture along with the alternate learning is responsible for a high computational load in our approach,

Figure X: Accuracy and feature importance ranking (FIR) scores yielded by different methods on the Enhancer-Promoter dataset: GM12878 Cell line (200 bp). While DFS and RF yield only one result with all the 102 features, RFE and ours produce the results at different subset sizes for $s = 15, 25, 35, 45, 55$. Note that the results yielded RFE and ours for $s = 35$ above are not specified deliberately with the subset size to indicate that those have been reported in the main text.

## C.2 Results on RNA-seq Data

Apart from the comparative study on the Enhancer-Promoter dataset, we have further applied our approach to the UCI gene expression cancer RNA-Seq data set[12], to evaluate our approach on a dataset of many features.

The gene expression cancer RNA-Seq dataset is part of The Cancer Genome Atlas Pan-cancer Analysis Project. The original data set is maintained by the cancer genome atlas pan-cancer analysis project. Gene expression data are composed of DNA microarray and RNA-seq data. Therefore, microarray data analysis facilitates the clarification of biological mechanisms and development of drugs toward a more predictable future. In comparison to hybridization-based microarray technology, RNA-seq has a larger range of expression levels and contains more information. RNA-Seq is a random extraction of gene expression of patients with five different types of tumors including BRCA (breast), KIRC (kidney), COAD (colon), LUAD (lung) and PRAD (prostate). The dataset contains 801 samples, each of which has 20,531 biological features or genes. The data set is imbalanced and there are 300, 146, 78, 141 and 136 samples for BRCA, KIRC, COAD, LUAD and PRAD, respectively.

Unlike other methods, e.g., [19], we do not pre-process the imbalanced data in our experiment apart from removal of 267 constant features. In other words, we use only 20,264 features in each sample to train our dual-net model. All the data are standardised with zero mean and unit standard deviation. The dataset is randomly split into two subsets, training and test, where there are 600 and 201 samples in the training and the test subset. The four-fold cross-validation working on the training subset are used for parameter estimate and hyperparameter tuning for our dual-net model. To make a fair comparison to the best performer on this dataset as reported in [19], we use the identical setting by using $s = 49$ in our experiment. The information on the dual-net architecture and optimal hyperparameter values used in this experiment is provided in Tables I and II, respectively.

Table V: Accuracy yielded by different methods on RNA-seq dataset (adapted from Table 7 in [19]).

| Method | # Samples | # Features | # Classes | # Selected Features | # Classifiers | Accuracy |
|---|---|---|---|---|---|---|
| [20] | 96 | 4026 | 9 | <60 | 1 | 0.9730 |
| [21] | 62 | 6000 | 2 | 15 | 1 | 0.9677 |
| [22] | 97 | 24481 | 2 | 7 | 7 | 0.9381 |
| [22] | 102 | 12600 | 2 | 4 | 7 | 0.9706 |
| [23] | 175 | 1072 | 2 | - | 110 | 0.9500 |
| [24] | 215 | 1047 | 4 | - | 20 | 0.9860 |
| [25] | 569 | 32 | 2 | 24 | 1 | 0.9877 |
| [19] | 801 | 20531 | 5 | 49 | 5 | 0.9881 |
| **Ours** | **801** | **20531** | **5** | **49** | **1** | **0.9938** |

Table V shows the existing results of several feature selection methods [19, 20, 21, 22, 23, 24, 25] on this dataset with various settings although most of the existing methods do not work on the entire dataset. From Table V, it is clearly seen that, under the same settings, our approach outperforms the best performer on this dataset in literature. The experimental result on this nontrivial real-world dataset demonstrates that our approach works well for a data set of many features as long as there are enough training examples required by deep learning. Thus, we firmly believe that our approach will be applicable to a large data set of many features, e.g., images where there are millions of pixels. We shall look into the scalability of our approach in our ongoing work.

In summary, our experimental results manifest that leveraged with deep learning, our approach outperforms a number of state-the-art FIR and feature selection methods on two biological datasets. This suggests that our approach would be a strong candidate for feature selection and feature importance ranking in real-world biological applications.

# D Pseudo Code

In this section, we describe the implementation of our alternate learning algorithm used to train our proposed dual-net neural architecture for feature importance ranking underlying feature selection. The pseudo code[13] in Algorithm 1 carries out the alternate learning algorithm as described in Sect. 3.3 of the main text. The pseudo code in Algorithm 2 implements a subroutine used in line 10 of Algorithm 1 to generate an optimal feature subset in the current step as described in Phase II.A (c.f. Sect. 3.3 in the main text). In Algorithm 1, lines 1-7 corresponds to Phase I, lines 9-12 carry out Phase II.A and lines 13-18 implement Phase II.B. Phase II.A and II.B alternate until the early stopping condition is satisfied as implemented by the loop from line 8 to line 23.

---

**Algorithm 1** Alternate Learning Algorithm

---

**Require:** loss function of operator net, $l(\boldsymbol{x} \otimes \boldsymbol{m}, \boldsymbol{y}; \theta)$, selector net, $f_S(\varphi, \boldsymbol{m})$
**Require:** feature set/subset size $d$ and $s$, fraction of random masks $f$, perturbation factor $s_p$
**Require:** number of optimal subset candidates $|\mathcal{M}'|$, number of batches $E_1$ in Phase I.
**Require:** mask weight vector used in the weighted selector loss $\boldsymbol{w}_S$ of $|\mathcal{M}'|$ elements.

1: **for** $e \leftarrow 0$ to $E_1$ **do**
2:      $\mathcal{M}'_1 = \{\boldsymbol{m}_i | \boldsymbol{m}_i = \text{Random}(\mathcal{M}, s)\}_{i=1}^{|\mathcal{M}'|}$         ▷ create a random batch of masks
3:      $\mathcal{L}_O(\mathcal{D}, \mathcal{M}'_1; \theta) = \frac{1}{|\mathcal{M}'||\mathcal{D}|} \sum_{\boldsymbol{m} \in \mathcal{M}'} \sum_{(\boldsymbol{x}, \boldsymbol{y}) \in \mathcal{D}} l(\boldsymbol{x} \otimes \boldsymbol{m}, \boldsymbol{y}; \theta)$      ▷ calculate operator loss
4:      $\theta'' \triangleq \theta' - \eta \nabla_\theta \mathcal{L}_O(\mathcal{D}, \mathcal{M}'_1; \theta)|_{\theta=\theta'}$         ▷ update the parameters in operator
5:      $\mathcal{L}_S(\mathcal{M}'; \varphi) = \frac{1}{2|\mathcal{M}'|} \sum_{\boldsymbol{m} \in \mathcal{M}'} \left( f_S(\varphi; \boldsymbol{m}) - \frac{1}{|\mathcal{D}|} \sum_{(\boldsymbol{x}, \boldsymbol{y}) \in \mathcal{D}} l(\boldsymbol{x} \otimes \boldsymbol{m}, \boldsymbol{y}; \theta) \right)^2$ ▷ calculate MSE loss of selector net
6:      $\varphi'' \triangleq \varphi' - \eta \nabla_\varphi \mathcal{L}_S(\mathcal{M}'_1; \varphi)|_{\varphi=\varphi'}$         ▷ update the weights in selector
7: **end for**
8: **for** $t \leftarrow 0$ to inf **do**
9:      $\mathcal{M}'_{t+1,1} = \{\boldsymbol{m}_i | \boldsymbol{m}_i = \text{Random}(\mathcal{M}, s)\}_{i=1}^{(1-f)|\mathcal{M}'|}$         ▷ create a random batch of masks
10:     $m_{t+1,opt} \Leftarrow generateOptimalMask(f_{SN}(\varphi_t))$         ▷ implemented in Algorithm 2
11:     $\mathcal{M}'_{t+1,2} = \{\boldsymbol{m}_{t,best}\} \cup \{\boldsymbol{m}_{t+1,opt}\} \cup \{\boldsymbol{m}_i | \boldsymbol{m}_i = \text{Perturb}(\boldsymbol{m}_{t+1,opt}, s_p)\}_{i=1}^{f|\mathcal{M}'|-2}$   ▷ collect the best mask from last step, the current optimal mask and those perturbed optimal masks
12:     $\mathcal{M}'_{t+1} = \mathcal{M}'_{t+1,1} \cup \mathcal{M}'_{t+1,2}$         ▷ form new subset candidates for operator
13:     $\mathcal{L}_O(\mathcal{D}, \mathcal{M}'_1; \theta) = \frac{1}{|\mathcal{M}'||\mathcal{D}|} \sum_{\boldsymbol{m} \in \mathcal{M}'} \sum_{(\boldsymbol{x}, \boldsymbol{y}) \in \mathcal{D}} l(\boldsymbol{x} \otimes \boldsymbol{m}, \boldsymbol{y}; \theta_t)$      ▷ calculate operator loss
14:     $\theta_{t+1} \triangleq \theta_t - \eta \nabla_\theta \mathcal{L}_O(\mathcal{D}, \mathcal{M}'_1; \theta)|_{\theta=\theta}$         ▷ update parameters in operator
15:     $\mathcal{L}_S(\mathcal{M}'; \varphi) = \frac{1}{2|\mathcal{M}'|} \sum_{i=0}^{|\mathcal{M}'|} w_{S,i} \left( f_S(\varphi; \boldsymbol{m}) - \frac{1}{|\mathcal{D}|} \sum_{(\boldsymbol{x}, \boldsymbol{y}) \in \mathcal{D}} l(\boldsymbol{x} \otimes \boldsymbol{m}, \boldsymbol{y}; \theta) \right)^2$ ▷ calculate the weighted MSE loss of selector
16:     $\varphi_{t+1} \triangleq \varphi_t - \eta \nabla_\varphi \mathcal{L}_S(\mathcal{M}'_1; \varphi)|_{\varphi=\varphi_t}$         ▷ update the parameters in selector
17:     $\boldsymbol{m}_{t+1,best} = \text{argmin}_{\boldsymbol{m} \in \mathcal{M}'_{t+1}} \left( \sum_{(\boldsymbol{x}, \boldsymbol{y}) \in \mathcal{D}} l(\boldsymbol{x} \otimes \boldsymbol{m}, \boldsymbol{y}; \theta) \right)$   ▷ record the best performed mask
18:     $\mathcal{L}_{t,m_{opt}} \leftarrow \left( \sum_{(\boldsymbol{x}, \boldsymbol{y}) \in \mathcal{D}} l(\boldsymbol{x} \otimes \boldsymbol{m}, \boldsymbol{y}; \theta) \right) [(1-f)+1]$   ▷ record loss of the $\boldsymbol{m}_{opt}$, $(1-f)+1$ should be its index
19:     **if** $checkEarlyStopping(\mathcal{L}_{t,m_{opt}})$ **then**
20:        $\theta_t = restoreBestWeights()$         ▷ stopping condition is met
21:        **break**
22:     **end if**
23: **end for**

**Algorithm 2** Generation of Optimal Feature Subset

---

**Require:** selector net $f_S(\varphi, \boldsymbol{m})$
**Require:** input feature set size $d$, selected subset size $s$

1: $\boldsymbol{m}_0 \leftarrow (\frac{1}{2}, \frac{1}{2}, ..., \frac{1}{2})$
2: $\boldsymbol{\delta}_{m_0} = \frac{\partial f_S(\varphi, \boldsymbol{m})}{\partial \boldsymbol{m}}|_{\boldsymbol{m}=\boldsymbol{m}_0}$              ▷ calculate input gradient
3: $\boldsymbol{m}_{opt} \leftarrow (0, 0, ..., 0)$
4: $(\boldsymbol{i}_{unmasked}, \boldsymbol{i}_{masked}) \leftarrow \text{argsort}(\boldsymbol{\delta}_{m_0})$   ▷ determine indexes with 1s (unmasked, top $s$ biggest gradients) and 0s (masked, the smallest gradients)
5: $\boldsymbol{m}_{opt}[\boldsymbol{i}_{unmasked}] \leftarrow (1, 1, ..., 1)$             ▷ complete creating $\boldsymbol{m}_{opt}$
6: $\boldsymbol{\delta}_{m_{opt}} = \frac{\partial f_{SN}(\varphi, \boldsymbol{m})}{\partial \boldsymbol{m}}|_{\boldsymbol{m}=\boldsymbol{m}_{opt}}$        ▷ reclalculate the gradient at $\boldsymbol{m} = \boldsymbol{m}_{opt}$
7: $i_{min} \leftarrow \text{argmin}(\boldsymbol{\delta}_{m_{opt}}[i_{unmasked}])$       ▷ index of minimum unmasked gradient
8: $i_{max} \leftarrow \text{argmax}(\boldsymbol{\delta}_{m_{opt}}[i_{masked}])$         ▷ index of maximum masked gradient
9: $\boldsymbol{i}_{neg} \leftarrow \text{argwhere}(\boldsymbol{\delta}_{m_{opt}}[i_{unmasked}] < 0)$ ▷ create a set of unmasked indices that have negative gradients
10: **for** $i$ in $\boldsymbol{i}_{neg}$ **do**
11:     $\boldsymbol{m}'_{opt} \leftarrow \boldsymbol{m}_{opt}$                      ▷ **Validation step 1**
12:     $\boldsymbol{m}'_{opt}[i] \leftarrow 0$          ▷ mask the negative, previously unmasked, input
13:     $\boldsymbol{m}'_{opt}[i_{max}] \leftarrow 1$      ▷ unmask the biggest (gradient-wise), previously masked input
14:     **if** $f_S(\varphi, \boldsymbol{m}''_{opt}) < f_S(\varphi, \boldsymbol{m}_{opt})$ **then**
15:        $\boldsymbol{m}_{opt} \leftarrow \boldsymbol{m}'_{opt}$          ▷ replace $\boldsymbol{m}_{opt}$ and restart the validation
16:        recalcualte $\boldsymbol{i}_{unmasked}$ and $\boldsymbol{i}_{masked}$
17:        goto step 6
18:     **end if**
19: **end for**
20: $\boldsymbol{m}''_{opt} \leftarrow \boldsymbol{m}_{opt}$                       ▷ **Validation step 2**
21: $\boldsymbol{m}''_{opt}[i_{min}] \leftarrow 0$      ▷ mask the smallest (gradient-wise), previously unmasked, input
22: $\boldsymbol{m}''_{opt}[i_{max}] \leftarrow 1$      ▷ unmask the biggest (gradient-wise), previously masked input
23: **if** $f_S(\varphi, \boldsymbol{m}''_{opt}) < f_{SN}(\varphi, \boldsymbol{m}_{opt})$ **then**
24:     $\boldsymbol{m}_{opt} \leftarrow \boldsymbol{m}''_{opt}$          ▷ replace $\boldsymbol{m}_{opt}$ and restart the validation
25:     recalculate $\boldsymbol{i}_{masked}$ and $\boldsymbol{i}_{masked}$
26:     goto step 6
27: **end if**

---

## Footnotes

[1]In our alternate learning procedure, we use the operator loss incurred by the optimal subset, $\boldsymbol{m}_{t,opt}$, on the validation set for early stopping. For clarity and details, see Section B.1.

[2]BAHSIC webpage: https://www.cc.gatech.edu/~lsong/code.html

[3]PyMRMR library: https://pypi.org/project/pymrmr/

[4]CCM repository: https://github.com/Jianbo-Lab/CCM

[5][online]: https://archive.ics.uci.edu/ml/machine-learning-databases/glass/

[6][online]: https://www.openml.org/t/3815

[7][online]: https://www.openml.org/d/307

[8][online]: http://featureselection.asu.edu/old/datasets.php

[9][online]: http://www.cad.zju.edu.cn/home/dengcai/Data/FaceData.html

[10][online]: `https://github.com/yifeng-li/DECRES/tree/master/data`

[11]We do this experiment with the authors' code: https://gitlab.com/skblaz/attentionrank.

[12][online]: https://archive.ics.uci.edu/ml/datasets/gene+expression+cancer+RNA-Seq

[13]Our source code and all the related information regarding the experimental settings are available online: https://github.com/maksym33/FeatureImportanceDL.