[Reviews · NeurIPS 2020]

Review 1

Summary and Contributions: This paper tackles the np-hard problem of feature selection (importance ranking) in the context of deep neural networks for tasks of classification and regression. In particular, they propose the instantiation of (a) operator network, that is optimised for the end task while using a subset of binary masks for feature selection as provided by the selector network, and (b) the selector network which uses the feedback of the operator, in terms of the performance at end task, to predict better/more suitable feature masks. More particularly, given a subset of binary masks, the operator network is adapted to maximise the performance at end task using these masks. The selector network is then calibrated, such that for each mask it outputs a score which corroborates with the operator's performance on it. Gradients of the scoring network on random or perturbed masks are then used to evolve an improved subset of masks, which is fed back into the operator, and the training of the two network continues in alternation. The contributions are such: a) A two-unit scheme for feature selection. b) Losses and scheme to train these units. c) Experimental analysis on an array of datasets to quantify their performance.

Strengths: ================Post rebuttal======================= Connection to the papers I recommended has been made clear (instance-wise versus population-wise) as is also the case for the suggestions made by R3. The authors have provided detailed contextualisation and agreed to add it to the paper. In terms of performance comparison, they have added another experiment to their evaluation with a much larger set of features, to demonstrate the scalability to large-sized datasets. From the experiments in the paper, and given the new addition, it becomes clear that their method is able to yield higher performance via the selection of smaller number of features. The computational cost is indeed high, given the alternate training routine involving two full-size DNNs, which going forward could be an interesting area to work in i.e. downscaling of these units. For now, the authors have agreed to add the computational cost analysis to the paper. As is also the case for the convergence plots, which like R3 mentioned, I would also like to see in the main text. All in all, I think with the evolutionary computation style formulation of their selection phase, and with the quality of results, this would be a good addition to the papers at neurips. ================================================= a ) This work is an empirical evaluation of a two part pipeline proposed for feature selection/ranking. Feature selection is a combinatorial problem, and one that is of crucial relevance to the machine learning community, through its association with an array of questions such as input denoising (suppression of irrelevant features), network generalisability and network understanding/explainability. Thus, the contributions of the authors in this domain are useful. b) The method seems to perform at par with existing methods, across an array of benchmark datasets. It also seems to favour smaller feature sets as compared to existing methods.

Weaknesses: a) The operator network is straightforward in construct and attempts to maximise the performance given the subset of binary masks. However, it may be biased towards features that work well independently, as opposed to those that independently perform poorly but combinations of which are useful for end task, especially for high dimensional inputs. In other words, it is biased towards linear solutions. b) The related work is not very well developed, for instance, the application of their method to visual data (MNIST) urges contextualisation amongst more recent methods such as Reinforce or continuous importance scoring such as in Learn To Pay Attention. c) The scheme of generating mask subsets in Phase II-A contains a series of adhoc steps, which could be contextualised better. The steps seem to bear semblance with evolutionary methods, with their use of a fitness function (similar to the output of the operator network), and mutations (similar to perturbations on masks). To this the authors have added the generation of more optimal masks by feature level swapping using the gradients of the selector/scoring network. Perhaps, there is a way to present their approach in a more organised way. d) The computational cost of their method is not detailed, and is likely to be high (only briefly mentioned in conclusion) given the alternate training routine. e) The performance is comparable to existing methods and the motivation for their complicated training procedure is not clear.

Correctness: Seems to be.

Clarity: It could be organised better. Please refer to the notes under weakness section.

Relation to Prior Work: The related work could be developed better. There are a variety of deep learning models in the context of visual tasks that have implemented this sort of feature selection such as Reinforce, or LTPA (real valued importance scores). How does their approach compare with those?

Reproducibility: Yes

Additional Feedback:


Review 2

Summary and Contributions: The paper proposes a Feature importance ranking framework for deep learning models. The proposed algorithm involves training of an additional network (selector net) along with training the original network, referred to as the operator net. The framework is empirically validated using multiple data sets - synthetic, benchmark and real data, and compared against numerous other FIR methods.

Strengths: 1. While interpretability, explainability in DL have been extensively studied over the past few years, pretty much all those studies focused on identifying important features in the input data, but for a specific test sample at a time. The authors referred to these methods as 'instance-wise' interpretability. On the other hand, global interpretability or 'population-wise' interpretability (as termed here) techniques have not been explored much. In this context, this paper is an addition to that sparse literature, that proposes a new algorithm for FIR in deep models. 2. The empirical evaluation is pretty comprehensive, with diverse data sets, multiple competing methods. In this kind study, synthetic data with known feature importance is critical to verify the effectiveness of an algorithm. Therefore, the demonstration of the proposed technique outperforming other methods for synthetic data is quite useful. Beyond that, results on benchmark data sets, real data sets are also fairly convincing.

Weaknesses: 1. A critical aspect of the proposed framework is that it involves an alternating minimization approach for simultaneous training of the operator net and the selector net. Some convergence analysis would be useful to convince the reader about the validity of this algorithm. The authors do discuss this matter empirically though in the supplementary section with loss curves for the two networks. At the very least, maybe some of these empirical observations can be moved to the main paper. 2. Discussion of the existing literature and the landscape of DL interpretability methods seem a bit incomplete and the specific contributions need to be clearly articulated. For example, although "population-wise" interpretability methods are not abundant, the authors do not discuss some of the recent works that I believe are relevant - https://arxiv.org/pdf/1907.03039.pdf, Also, there is the class of attention based models that researchers have used feature selection that would be worth considering (e.g., I saw a recent one - https://arxiv.org/pdf/2002.04464.pdf). 3. While comparison with 'instance-wise' methods could be seen as apples vs. oranges, I believe some comparison with important visual features identified here with those identified by 'instance-wise' methods would give us critical insights on how the global interpretability problem differs from the local interpretability methods. 4. Finally, discussions on computational overhead of the proposed algorithm and feasibility of applying the proposed scheme for large data sets (e.g., imagenet) should be discussed.

Correctness: Claims and methods (algorithm and empirical experiments) are correct.

Clarity: The paper is generally lucidly written. Although some clarity on exact contributions would be very useful as mentioned earlier.

Relation to Prior Work: This is discussed in detail in the weakness section.

Reproducibility: Yes

Additional Feedback: Post-rebuttal: I have read the authors' response. The authors acknowledge the general issues raised by myself and other reviewers regarding comparison with certain methods and computation issues. Unfortunately, it is not fair to expect such substantial additions in the revision phase. Therefore, I think the paper is not at a clear 'accept' stage. So, no change of score from my end.


Review 3

Summary and Contributions: In this paper, the authors propose a deep learning model with dual networks to solve the feature importance ranking problem. They train two networks, an “operator” and a “selector”, and they are trained jointly in an alternate manner. In each such phase, the “selector” network half-guesses half-learns the most important features, and the “operator” network in turn trains using only these chosen features. The authors show improved results over previous works in synthetic datasets, benchmark datasets and real-world datasets.

Strengths: The authors present a novel and interesting approach for dealing with feature importance ranking. The experiments presented show that their method achieves higher accuracy results while using fewer features than other methods.

Weaknesses: It seems that the major drawback of the method is the fact that the number of “important features”, denoted by s, must be chosen in advance. Different s values will probably yield very different results. In section A.1 of the supplementary, the authors note that the input dimension of the "operator" network was extended to be 2d rather than d. Was this also the case when comparing to the other FIR methods?

Correctness: The claims and method seem to be correct.

Clarity: The paper is written very clearly, although it relies heavily on one reading the supplementary (many important details are not included in the paper itself). Section 3.3 is somewhat hard to read because of all the enumeration within the text, but I assume this is due to lack of space. Please review the paper for several grammatical errors/typos.

Relation to Prior Work: While the paper lists several previous methods in the Related Work section, and also compares its experiments with theirs, it seems that an overview of how these previous methods work is not provided.

Reproducibility: Yes

Additional Feedback: A technical point: In equation 2b, why is the loss divided by 2?

[Author Response · NeurIPS 2020]

# Author Response

We thank three reviewers for their valuable feedback. We address the comments and the concerns as follows.

## To Reviewer #2

**a) Feature independence**. Our model does not work on the feature independence assumption but in a performance-driven manner. If a subset of combined features yields the optimal performance, the feature subset will be selected. Our experiments suggest that the importance of a feature correlated with others in the selected subset is ranked higher than that of an independent feature less relevant to the target. It is hence not biased towards linear solutions.

**b) Related work**. (1) REINFORCE and LTPA were proposed for instancewise FIR while ours tackles populationwise FIR problems. In fact, instancewise FIR for local explanations is quite distinct from populationwise FIR for global explanations and converting instancewise to populationwise FIR requires non-trivial mechanisms (see https://arxiv.org/pdf/1907.03039.pdf). Thus, we cannot directly compare ours to such instancewise methods. (2) Technically, LTPA does not work on input features and hence cannot conduct feature selection. REINFORCE works only for visual input while ours works for different input types including visual input. REINFORCE was implemented with reinforcement learning due to its non-differentiable nature while our model is carried out with supervised learning.

**c) Connection to evolutionary computation (EC)**. We really appreciate your insight by understanding our work from an EC perspective. This insight may allow us to highlight our contributions from another angle: for feature selection, (1) ours uses a single learning model (enabling different feature masks to be used simultaneously during learning) to carry out the functionality of a population of learning models in EC; (2) instead of purely stochastic operations on population in EC, ours uses a more efficient gradient-guided local stochastic search strategy. We will add this insight in revision.

**d) Computational cost**. Indeed, computational cost of our dual-net model is high due to the use of two DNNs and the alternate learning routine. We will re-run all experiments on the same environment and report detailed results.

**e) Performance and motivation**. (1) Our model works well for a large feature set when there are enough training examples required by deep learning. For demonstration, we have just conducted an experiment on the UCI gene expression cancer RNA-Seq data set (https://archive.ics.uci.edu/ml/datasets/gene+expression+cancer+RNA-Seq), where there are 801 instances and 20,531 features. With 4-fold cross-validation (450, 150 and 201 instances used for training, validation and test, respectively), our model yields $99.38 \pm 0.00\%$ testing accuracy on 49 selected features ($s = 49$). In literature, to the best of our knowledge, the best performance in the same settings on this data set is $98.81\%$ (see Highlight and Table 7, https://doi.org/10.1016/j.ygeno.2019.11.004). For your information, the code provided by the authors of CCM [14] (the strongest method in our comparative study) does not work on our server as it requires 158GB+ memory for this data set. (2) The motivation of our training procedure is generally described in paragraph 1 of Sect. 3.3 and the input-gradient guided local search idea was motivated by the work presented in [12], as stated in Phase II-A.

## To Reviewer #3

**1. Convergence analysis**. We agree to this point. We will summarize empirical observations in the revised main text and make a formal convergence analysis of our alternate learning algorithm in our ongoing research.

**2. Connection to latest work**. Thanks for pointing out two papers related to our work. Our work distinguishes those from the following aspects: a) those methods yield populationwise FIR by aggregating or integrating instancewise FIR in a sub-optimal manner, while our models directly learns populationwise FIR in an optimal way; b) for feature selection, those methods work as filtering so another learner has to be re-trained on the selected subset for a target task, while ours works as embedding by accomplishing feature selection and a target task together in an end-to-end manner. We will make a connection to two papers and report comparative results between ours and those methods in revision.

**3. Instancewise vs. populationwise**. We entirely agree to this point. We will be extending our model to instancewise FIR and study a connection between population and instancewise FIR in our ongoing work.

**4. Computational cost and scalability**. a) Regarding the computational overhead issue, see our response **d)** to reviewer #2; b) We will investigate scalability issues although any deep learning models effectively working for a large dataset can be used as an operator in our dual-net model.

## To Reviewer #4

**Number of important features**. Our problem formulation for a fixed $s$ is a common setting in feature selection, e.g., CCM [14]. When the ground-truth is unknown in real applications, results yielded by different $s$ values (sub-solutions to feature selection) are still useful to reveal some meaningful relationship between selected features and the target. To find out the optimal value of $s$, the common setting allows a model like ours to work on different $s$ values in parallel.

**Other questions**. a) in all the experiments, $2d$ is the input dimension of the operator in ours while $d$ is the input dimension of all other methods; b) In Eq.(2b), dividing 2 in the MSE loss is default to facilitate the gradient computation.

[Meta-Review · NeurIPS 2020]

This paper describes an approach for feature selection for deep learning based on two separate networks. All reviewers have found some issues, but tend to accepting the paper, and the authors have promised to make the necessary improvements in the presentation.